# Assessing the Diversity and Systematics of Brachyopini Hoverflies (Diptera: Syrphidae) in the Iberian Peninsula, Including the Descriptions of Two New Species [note 1]

**DOI:** 10.3390/insects13070648

**Published:** 2022-07-18

**Authors:** Antonio Ricarte, Zorica Nedeljković, Pablo Aguado-Aranda, Mª Ángeles Marcos-García

**Affiliations:** Research Institute CIBIO (Centro Iberoamericano de la Biodiversidad), Science Park, University of Alicante, Ctra. San Vicente del Raspeig s/n, 03690 San Vicente del Raspeig, Alicante, Spain; ricarte24@gmail.com (A.R.); zoricaned14@gmail.com (Z.N.); marcos@ua.es (M.Á.M.-G.)

**Keywords:** *Chrysogaster*, COI barcode, identification key, *Melanogaster*, *Orthonevra*, Spain, syrphids

## Abstract

**Simple Summary:**

Hoverflies are a diverse family of insects (6000+ species) providing multiple ecosystem services and direct benefits to society, for example, in pollination and pest control. The extent to which hoverflies can benefit society depends on how well their basics are known, i.e., taxonomy (classification), biology and ecology. Small and inconspicuous hoverflies such as those of the genera *Chrysogaster*, *Melanogaster*, *Lejogaster*, *Orthonevra* and *Riponnensia* (tribe Brachyopini) do not usually catch the attention of the general public neither of scientists, and their classification is often poorly known at least in areas of their range. So, the aim of the present work is to gain knowledge in the classification of hoverflies by studying the species of the five genera above-mentioned in the Iberian Peninsula, which has one of the highest biodiversity levels in Europe. We discovered and described two new species from Spain, promoted a subspecies to species and proposed other nomenclatural acts to stabilise these hoverflies’ classification in Europe. Our results are reinforced with DNA analysis to locate the studied species in a wider systematic framework within the Brachyopini.

**Abstract:**

Five genera of Brachyopini, *Chrysogaster* Meigen, 1800, *Melanogaster* Rondani, 1857, *Lejogaster* Rondani, 1857, *Orthonevra* Macquart, 1829 and *Riponnensia* Maibach et al. 1994a are here revised from the Iberian region. Two new species, *Melanogaster baetica* Ricarte and Nedeljković, sp. n. and *Orthonevra arcana* Ricarte and Nedeljković sp. n., are described from Spain, and a third species, *Chrysogaster coerulea* Strobl *in* Czerny and Strobl, 1909 stat. n., is reinstated as valid and redescribed. A lectotype is designated for *Orthonevra plumbago* (Loew, 1840). The holotype of *Orthonevra incisa* (Loew, 1843) and the lectotype of *O. plumbago* are described in detail and illustrated. *Melanogaster baetica* sp. n. is similar to *Melanogaster parumplicata* (Loew, 1840) in male genitalia morphology, while *O. arcana* sp. n. is similar to *O. incisa* in the entirely-pollinose sternum I and the conspicuous incision on the posterior margin of tergum V in female. The first Iberian record of *Chrysogaster rondanii* Maibach and Goeldlin de Tiefenau, 1995 is provided, whilst *Melanogaster aerosa* is removed from the Iberian checklist of Syrphidae. Identification keys are presented to the five Brachyopini genera and 18 species now reported from the Iberian Peninsula (*Chrysogaster*, 6 spp.; *Lejogaster*, 2 spp.; *Melanogaster*, 3 spp.; *Orthonevra*, 5 spp.; *Riponnensia*, 2 spp.). COI (Cytochrome c oxidase subunit I) barcodes of the two new species plus *C. coerulea*, *Chrysogaster solstitialis* (Fallén, 1817), *Orthonevra nobilis* (Fallén, 1817) and *Orthonevra frontalis* (Loew, 1843) were successfully obtained from Spanish specimens. A COI-based tree was produced to locate these taxa in a wider systematic framework within the tribe.

## 1. Introduction

The taxonomic concept of the Brachyopini Williston, 1885 (Syrphidae: Eristalinae) is complex and has changed along history. Hull [1] grouped under this tribe (as ‘Chrysogastrini’) flies with straight face, projecting epistoma, flattened abdomen, pubescent (pollinose) spots upon the face and a typical wing venation. Genera such as *Psilota* Meigen, 1822 were included in Hull’s concept of the brachyopines, while *Myolepta* Newman, 1838 was excluded. In his work on the Neotropical Milesinae, Thompson [2] defined the tribe (as ‘Chrysogasterini’) as flies with vein r-m located before the middle of the cell DM, bare eyes, concave face in females, bare metasternum and without subscutellar fringe, excluding genera such as *Psilota* but placing others such as *Myolepta* within. Thompson [2] himself recognised problems in his tribal concept, since certain species of *Myolepta* did not meet some of the defining character states of the tribe. 

Thompson [3] recovered the older tribal name, Brachyopini Williston, 1885, to refer to the chrysogasterines of authors and Thompson and Rotheray [4] followed it. In the meantime, Peck [5] still used the name Chrysogasterini in his Palaearctic catalogue of syrphids after Thompson [2] (but neglecting Thompson [3]), and then Maibach et al. [6,7] followed Peck [5]. Maibach et al. [6] described a new genus of brachyopines, *Riponnensia* Maibach et al. 1994a, that fitted with difficulty to Thompson’s tribal concept from 1972, as the new genus has a short disperse pilosity on the eyes and a pilose metasternum. Other authors such as Kassebeer [8] and Vujić [9] also used the name ‘Chrysogasterini’ extensively in their works. Kassebeer [10] and Moran et al. [11] used the name Brachyopini. Moran et al. [11] divided the Brachyopini further into the subtribes Brachyopina and Spheginina and, not unexpectedly, found the tribe split into six different clades far apart in their analyses of Eristalinae. Whatever the name is, all suprageneric classifications coincide in grouping the west-Palaearctic genera *Chrysogaster* Meigen, 1800, *Melanogaster* Rondani, 1857, *Lejogaster* Rondani, 1857, *Orthonevra* Macquart, 1829 and, later on, *Riponnensia* and *Ighboulomyia* Kassebeer, 1999b under the same tribe. All except *Ighboulomyia* are to be treated in the present paper, because *Ighboulomyia* is restricted to North Africa.

According to recent literature, 15 species of *Chrysogaster*, *Melanogaster*, *Lejogaster*, *Orthonevra* and *Riponnensia* are reported from the Iberian Peninsula: *Chrysogaster basalis* Loew, 1857, *Chrysogaster coemiteriorum* (Linnaeus, 1758), *Chrysogaster solstitialis* (Fallén, 1817), *Chrysogaster virescens* Loew, 1854, *Lejogaster metallina* (Fabricius, 1781), *Lejogaster tarsata* (Meigen, 1822), *Melanogaster aerosa* (Loew, 1843), *Melanogaster hirtella* (Loew, 1843), *Melanogaster nuda* (Macquart, 1829), *Orthonevra brevicornis* (Loew, 1843), *Orthonevra frontalis* (Loew, 1843), *Orthonevra nobilis* (Fallén, 1817), *Orthonevra onytes* (Séguy, 1961), *Riponnensia longicornis* (Loew, 1843), and *Riponnensia splendens* (Meigen, 1822) ([12,13,14]. The taxonomic status of some species in the west Palaearctic, for example, *Orthonevra tristis* (Loew, 1781) vs. *Orthonevra onytes* (Séguy, 1961), is still uncertain, as well as the actual systematic position of species such as *Orthonevra incisa* (Loew, 1843) [7]. However, relying on a more resolved generic framework [6,7], some taxonomic studies have been published describing new species from Europe [9,15] and North Africa [8,10].

Since the last and only monograph on the Syrphidae from Spain [16], the species of Brachyopini have yet to be revised again in the Iberian Peninsula. The main aim of the present paper is to update the knowledge on Brachyopini diversity in this geographic region and help stabilizing their species’ taxonomy in this part of the west Palaearctic.

## 2. Material and Methods

### 2.1. Examined Material

Specimens from different Spanish localities were examined at the following collections: University of Alicante Entomological Collection (CEUA-CIBIO, Alicante, Spain); Museo Nacional de Ciencias Naturales (MNCN, Madrid, Spain); University of Murcia zoological collection (UMCZ, Murcia, Spain); private collection of Miguel Carles-Tolrá (MCT, Barcelona, Spain). Material from the Museum für Naturkunde Berlin (ZMB, Berlin, Germany) and the Bergen Museum, University of Bergen (ZMBN, Bergen, Norway) was also examined. The collection acronym is indicated, together with a unique specimen code, after the number of males/females examined from each locality/date/collector. Examined material from Spain was as old as 119 years, but also included specimens recently collected by the authors of the present paper in different fieldwork campaigns in 2019–2021. Actual specimens were examined alongside literature on Iberian Brachyopini. For the examined material, novel data are indicated as ‘new’, and data on specimens published prior to the present study as ‘revised’. For non-examined material, published references are provided as ‘additional published records’.

### 2.2. Morphological Study

Species morphology was studied with a binocular microscope Leica M80. For descriptions, the following parameters were measured on one or more (‘*n*’) of the studied adult specimens: (a) body length (L), measured from the antennal insertions to the tip of the abdomen; (b) wing length (WL), measured from the tegula to the wing apex; (c) length/width ratio of the basoflagellomere, measured as the width at its maximum and the length from the most distal point of the pedicel to the basoflagellomere apex. In the species descriptions/redescriptions, measurements of the holotype/neotype are given, as well as a range of measurements from a number of specimens (*n*) in brackets. Measurements were taken with the software Leica Application Suite X (LAS X) ^®^ (v. 3.0.4.16529) from a specimen image made with a camera (Leica DFC 450) attached to a binocular microscope (Leica M205 C). Adult images were created as stacks of photos with the same equipment and software. Male genitalia were dissected in most cases for species confirmation and comparison. Genitalia were dissected following Ricarte et al. [17], but usually increasing the boiling time up to 10 min due to their high level of sclerotization. Male genitalia were drawn from individual images taken with the above-mentioned equipment and traced on a light table. Morphological terminology follows Thompson [18], except for the term ‘hair/s’, which is used in replacement of ‘pilis/pile’. For the male genitalia, terminology followed Maibach et al. [6]. For the illustrations cited from other publications, ‘figure’ is used in the text after a literature citation in square brackets, while for those original from the present work ‘Figure’ is used elsewhere.

### 2.3. Molecular Study

DNA was extracted both from one mesoleg and one metaleg of three males of *Chrysogaster coerulea* Strobl *in* Czerny and Strobl (1909), eight specimens of *C. solstitialis* (four males and four females), five paratypes of the new *Melanogaster* species (three males and two females), three paratypes of the new *Orthonevra* species (one male and two females), two specimens of *O. frontalis* (one male and one female), two males of *O. nobilis* and two males of *R. splendens* using the NZY Tissue gDNA Isolation kit following the manufacture’s protocol for animal tissues. PCR amplifications of the 5′ end of the Cytochrome *c* oxidase subunit I (COI-5′) gene were performed with the universal primers LCO1490 (5′-GGTCAACAAATCATAAAGATATTGG-3′) and HCO2198 (5′-TAAACTTCAGGGTGACCAAAAAATCA-3′) [19]. All amplifications were carried out in a total volume of 25 µL containing 1× of Buffer reaction, 0.4 mM of dNTPs, 0.2 µM of each primer, 2 mM of MgCl_2_ and 1 unit of NZYTaq II DNA polymerase. Thermocycler conditions followed Ståhls and Barkalov [20] but annealing at 47–49 °C. PCR products were visualized with an electrophoresis process in a 1% agarose gel and sequenced at Macrogen Inc. (Macrogen, Madrid, Spain).

The COI-5′ barcode sequences obtained were edited by eye with the program Sequencher v.5.4.6 (Gene Codes Corporation 2017, Ann Arbor, MI, USA). Then, COI-5′ sequences of each species of *Chrysogaster* Meigen, 1803, *Lejogaster* Rondani, 1857, *Melanogaster* Rondani, 1857 and *Orthonevra* Macquart, 1829 available at the public repositories GenBank and BOLD [21] were downloaded (see Table 1). For the molecular analyses, we excluded barcodes of specimens non-identified at the species level. First, alignment was performed automatically with MAFFT online service [22], which was checked by eye with the program AliView v.1.25 [23]. The final matrix had a length of 658 bp. Neighbour-joining (NJ) and maximum likelihood (ML) analyses were carried out in MEGA7 [24] with 1000 bootstrap replications. The NJ analysis was performed using the maximum composite likelihood model [25] and the ML analysis with the general time reversible [26] model with a gamma distribution (+G). This was the second evolutionary model proposed by the Corrected Akaike Information Criterion (AICc, 3749.85)—the first model was TIM2+I+G (3735.25)—using the software jModelTest v.2.1.10 [27]. The resulting trees were rooted based on a *Ceriana vespiformis* (Latreille, 1809) sequence. Additionally, the evolutionary divergence between barcode sequences was also estimated using the maximum composite likelihood model with a gamma distribution (0.168) in MEGA7 (see Appendix A).

## 3. Results

### 3.1. New Brachyopini Species

***Melanogaster baetica*****Ricarte and Nedeljković, sp. n.** (Figure 1, Figure 2A,B, and Figure 3)

Holotype

SPAIN • ♂; Jaén, Sierra de Cazorla, Ca. Arroyo Frío; 31 May 1993; Col. C.M. Herrera; *Cistus monspeliensis*; CEUA-CIBIO, CEUA00106948.

Paratypes

SPAIN • 1 ♂; same data as for the holotype; CEUA-CIBIO, CEUA00106925 • 1 ♂; Alicante, Alcoy, Font Roja; 12 April 1994; P. M. Isidro and C. Pérez Bañón leg.; ‘556’ on the label reverse; CEUA-CIBIO, CEUA00107229 • 1 ♂; Jaén, P. N. Cazorla, Vadillo Castril, Arroyo Tornillos de Gualay; 37°52′2″ N, 2°55′35″ O; 1575 m; 14 June 2019; Antonio Ricarte leg.; INV09538, DNA CEUA_S73; CEUA-CIBIO, CEUA00107228 • 1 ♀; same data as for the holotype; CEUA-CIBIO, CEUA00106954 • 1 ♀; Granada, Sierra Nevada, Güejar Sierra, El Dornajo, Parking camino a Peña del Perro; 1895 m; 24 June 2021; Z. Nedeljković leg.; CEUA-CIBIO, CEUA00109671 • 1 ♂; same data as for preceding; I. Ballester Torres leg.; CEUA-CIBIO, CEUA00109673 • 1 ♀; same data as for preceding; CEUA_S100; CEUA-CIBIO, CEUA00109543 • 2 ♂♂, 3 ♀♀; same data as for preceding; A. Ricarte leg.; CEUA_S101; CEUA-CIBIO, CEUA00109544, CEUA00109546 to 00109548, CEUA00109672 • 2 ♂♂; same locality as for preceding; 26 June 2021; I. Ballester Torres leg.; CEUA_S98 to S99; CEUA-CIBIO, CEUA00109540 to 00109541 • 1 ♀; same data as for preceding; Z. Nedeljković leg.; CEUA00109542 • 1 ♂; same data as for preceding; A. Ricarte leg.; CEUA-CIBIO, CEUA00109545 • 1 ♂; Valencia, Chelva; 10 May 1994; C. Pérez Bañón leg.; CEUA-CIBIO, CEUA00026277. The holotype and all male paratypes have the genitalia dissected and stored in a plastic microvial. The right legs are missing in the male from 2019.

*Examined material of similar* Melanogaster *species*. *Melanogaster aerosa*. NORWAY • 1 ♂; Figgjo, Ry: Høyland; 14 August 1965; Tore Nielsen leg.; ZMBN, ZMBN Dip—5832; A. Maibach February 2003 det. as *Melanogaster aerosa*; genitalia dissected and stored in a plastic microvial • 1 ♂; Figgjo, Ry: Høyland; 28 July 1962; Tore Nielsen leg.; ZMBN, ZMBN Dip—5831; A. Maibach February 2003 det. as *Melanogaster aerosa*; genitalia dissected and stored in a plastic microvial • 1 ♂; Kronåsen, HOy: Fana; 7 June 1970; Tore Nielsen leg.; ZMBN, ZMBN Dip—5838; A. Maibach February 2003 det. as *Melanogaster aerosa*; genitalia dissected and stored in a plastic microvial • 1 ♂; Kleppe, Ry: Klepp; 16 July 1965; Tore Nielsen leg.; ZMBN, ZMBN Dip—5834; Tore Nielsen det. as *Chrysogaster macquarti*; A. Maibach February 2003 det. as *Melanogaster aerosa* • 1 ♀; Gimra, Ry: Sola; 13 August 1965; Tore Nielsen leg.; *Sanguisorba officinalis* L.; ZMBN, ZMBN Dip—5827; Tore Nielsen det. as *Chrysogaster macquarti*; A. Maibach February 2003 det. as *Melanogaster aerosa* • 1 ♀; Figgjo, Ry: Høyland; 9 August 1960; Tore Nielsen leg.; ZMBN, ZMBN Dip—5829; Tore Nielsen det. as *Chrysogaster macquarti*; A. Maibach February 2003 det. as *Melanogaster aerosa*. *Melanogaster parumplicata*. NORWAY • 1 ♂; Miravatn, HOy: Fana; 11 June 1968; Tore Nielsen leg.; ZMBN, ZMBN Dip—5883; A. Maibach February 2003 det. as *Melanogaster parumplicata*; published in [7] • 1 ♂; Paradis, Hoy: Fana; 15 June 1968; Tore Nielsen leg.; ZMBN, ZMBN Dip—5881; A. Maibach February 2003 det. as *Melanogaster parumplicata*; genitalia dissected and stored in a plastic microvial • 1 ♀; same data as for preceding; ZMBN, ZMBN Dip—5880; A. Maibach February 2003 det. as *Melanogaster parumplicata* • 1 ♀; Åstveit, HOy: Åsane; 30 May 1971; Tore Nielsen leg.; ZMBN, ZMBN Dip—5888; A. Maibach February 2003 det. as *Melanogaster parumplicata* • GERMANY • 1 ♂; no further data; ZFMK-TIS-2608663 • 1 ♂; Schleswig-Holstein, Kreis Ostholstein, Glinde sudlich Schönwalde am Bungberg; ZFMK-TIS-2608582; C. Kassebeer det.

*Diagnosis*. This species can be distinguished from all other *Melanogaster* species by the following combination of features: black body with blue reflections (Figure 1); basoflagellomere roundish (Figure 2A), 0.7–0.9× longer than wide; vertical triangle with long black to dark brown hairs (Figure 2A); eye contiguity half the length of frontal triangle in dorsal view (males); in male, face with a slight but conspicuous tubercle centrally (Figure 2A); in female, occiput wholly white pilose, males with some black hairs; wing with a conspicuous large brown pigmented area anteriorly (Figure 1B,D); scutum with long erect white to yellowish white hairs, and a few long black hairs intermixed (Figure 2B); in males, abdomen with subparallel lateral sides (Figure 1B); in females, face flat, without tubercle (Figure 1C); surstylus of male genitalia, in lateral view, expanded subapically and then tapering abruptly in a short apex (Figure 3A).

*Description*. MALE (*holotype*). *Overall appearance* (Figure 1). Black body with blue reflections; L = 7 mm (6.3–7.1 mm, *n* = 4); WL = 5.6 mm (5.2–5.6 mm, *n* = 4). *Head* (Figure 2A). Eye with short sparse hairs; scape black; pedicel black, turning brownish black at the apex, with a few hairs ventrally and dorsally; basoflagellomere brownish black; arista longer than antenna (nearly 1.5×), dark brown, bare; basoflagellomere roundish (Figure 2A), 0.9× longer than wide (0.7–0.9×, *n* = 4); ocellar triangle conspicuously elevated, equilateral; vertical triangle with long black to dark brown hairs (Figure 2A); eye contiguity slightly longer than the vertical triangle, but half the length of frontal triangle in dorsal view; frontal triangle swollen (Figure 2A), shiny, coarsely punctured, with a slight medial excavation on the lower part; frontal triangle with long (as long as an antenna), black to dark brown hairs, shorter towards the antennae; lunule shiny, brownish black; face with a fascia of brown sparse pollinosity just below antennal insertions, from eye to eye; face finely wrinkled laterally, with a small but conspicuous tubercle centrally; face with white to brown hairs, except for the bare area below antennal insertions and tubercle; facial hairs shorter near the mouth; occiput with very sparse pollinosity, much denser on the eye margin; occiput with short white hairs, longer and black to brown dorsally. *Thorax*. Wing wholly microtrichose, with veins black to dark brown, except for orange basal part of Sc and CuA; pterostigma brownish black; wing with a conspicuous large brown pigmented area antero-centrally (Figure 1B,D); hairs on vein C black; vein M_1_ non-recessive (Figure 1B,D); squama yellowish white; halter with pedicel orange, but blackish base and capitulum; legs black, slightly lighter at the apices of meso- and metatibiae; leg hairs white to light yellow, except for the black hairs on protibia, ventral part of metafemur, and the setulae at the apex of mesotibia and tarsi (also some black hairs on tarsi dorsally); leg hairs longer on the postero-ventral part of pro- and mesofemora, with individual hairs wavy apically; legs sparsely pollinose; scutum shiny, sparsely (and inconspicuously) pollinose anteriorly and laterally; scutum with long erect white to yellowish white hairs (Figure 2B), and a few long black hairs intermixed; scutellum with both long and short white hairs intermixed; some long hairs of mesonotum with wavy apices; scutellum with a posterior slight sulcus; pleuron with white to light yellow hairs, except for the bare anterior anepisternum (ventrally with a small patch of hairs), dorso-medial and posterior anepimeron, katepimeron, meron and metasternum; katepisternum with a large bare area between the dorsal and ventral patches of hairs; anterior spiracle light orange; posterior spiracle dark brown. *Abdomen*. Elongated, with subparallel lateral sides (Figure 1B); in dorsal view, narrower than scutum at the maximum scutum width; thickly pollinose (dull) except for shiny lateral margins of terga II–IV, lateral margin of tergum I pollinose anteriorly; terga with white to light yellow hairs, short and recumbent centrally, conspicuously longer laterally, especially on the anterior corners of tergum II; all sterna shiny except for the entirely pollinose sternum I; sterna with white hairs, long and erect on sternum II, somewhat shorter and semi-recumbent on sternum III, recumbent on sternum IV; genitalia as in Figure 3, with surstylus more or less expanded subapically and then tapering abruptly in a short stocky apex, and hypandrium with a blunt projection on the outer side; superior lobe of the aedeagus cover with two opposite teeth of different size, the inner tooth much longer than the outer.

FEMALE (Figure 1C,D). Same as male except for the following: L = 6.8 mm; WL = 5.4 mm; basoflagellomere 0.8× longer than wide, orangey brown; frons with white to yellowish white hairs, as long as scape or nearly so, just slightly longer and forwardly inclined on the ocellar triangle; profile of the frons angled near antennae, i.e., with an abrupt change along the profile line of the frons near antennae in lateral view (Figure 1C); posterior part of frons with a differentiated medial sulcus bearing less hair density than contiguous parts of the frons; anterior part of frons (posterior to the frons angle) with slight lateral wrinkles diagonal to eye margin; face flat, without tubercle; face with pollinose triangular fascia; head without black hairs; apices of all femora narrowly brown; tarsi somewhat lighter than male; mesonotum with shorter hairs than male, as long as pedicel; abdomen roundish oval (Figure 1D), slightly wider than scutum at the maximum scutum width; tergum IV pollinose only on the anterior margin; tergum V entirely shiny; hairs of terga following the same pattern as in male, but of shorter length in general; sterna with shorter hairs than male, all semi-recumbent; sterna III and IV pollinose on the anterior margins.

*Etymology*. The specific epithet ‘baetica’, meaning Baetic in Latin, refers to the Spanish mountain range where most specimens of this species have been collected, the Baetic System.

*Range*. Spain (provinces of Alicante, Granada and Jaén).

*Biology*. Adults visit flowers of *Cistus monspeliensis* L. (Cistaceae) and *Euphorbia nicaeensis* All. (Euphorbiaceae). They can be found near streams (e.g., in ‘Tornillos de Gualay’) where their larvae are suspected to find their breeding sites. Adult records are from mid-April to late June. In the south of Spain, this species has been found from 1575–1895 m asl.

*Taxonomic notes*. This species is most similar to *M. parumplicata*, from which it can be distinguished by genitalic characters in males. For example, in the lateral view, (a) the surstylus of the new species is more or less expanded subapically and then tapers abruptly in a short stocky apex (Figure 3A), whilst in *M. parumplicata*, the subapical expansion is absent [9] (figure 21); (b) the projection of the outer side of the hypandrium is blunter in *M. baetica* sp. n. (Figure 3A) than in *M. parumplicata* [9] (figure 21). In addition, the long hairs of the scutum are mostly white to yellowish in males of *M. baetica* sp. n. but mostly dark brown to black in *M. parumplicata* (Figure 2D). As for the male (Figure 2A,C), the female of *M. baetica* sp. n. (Figure 1C) is similar to that of *M. parumplicata* in the face profile [7], but the female of *M. baetica* sp. n. has only white hairs on the occiput, whilst the examined females of *M. parumplicata* have at least some black hairs or setulae on the occiput. There is also a difference in the pollinose facial band in females. In *M. parumplicata*, this band is narrow laterally, while centrally, it is expanded towards the mouth, but in *M. baetica* sp. n., this fascia is wider laterally, in the shape of an inverted triangle.

The North African species *Melanogaster lindbergi* Kassebeer, 1999a is also similar to *M. baetica* sp. n., but they can be easily separated by the shape of the surstylus and the superior lobe of the aedeagus cover. The surstylus of *M. lindbergi*, in lateral view, is clearly notched subapically [8] (figure 3), whilst in *M. baetica* sp. n., it lacks this notch (Figure 3A). In addition, *M. lindbergi* has two opposite teeth of about the same size in the superior lobe of the aedeagus cover [8] (figure 3), but *M. baetica* sp. n. has a much longer tooth pointing towards the inner side of the hypandrium (Figure 3B). In the similar *M. aerosa*, the superior lobe of the aedeagus cover has a tooth pointing towards the outer side of the hypandrium, and this tooth is clearly longer than the one on the inner side [7] (figure 4), just opposite than in *M. baetica* sp. n. (Figure 3A). In general, both sexes of *M. baetica* sp. n. can be separated from *M. aerosa*, *M. lindbergi* and *M. parumplicata* by the blue reflections present, at least, on the thorax (Figure 1). For specimens of *M. baetica* sp. n. with darker reflections, the above-mentioned characters should be used for species separation.

***Orthonevra arcana* Ricarte and Nedeljković, sp. n.** (Figure 4, Figure 5, Figure 6 and Figure 7)

Holotype

SPAIN • ♂; Ciudad Real, Retuerta del Bullaque, el Chorrillo y cercanías; 28 February 2021; Z. Nedeljković leg.; flores *Salix atrocinerea*; CEUA-CIBIO, CEUA00107963.

Paratypes

SPAIN • 1 ♀; Escorial, Puerto; Colección Lauffer; MNCN, MNCN_Ent 302230 • 1 ♂; Ciudad Real, P. N. Cabañeros; 18 March–12 April 2005; A. Ricarte leg.; malaise trap maR1, 6898, Syrphidae, genitalia missing; CEUA-CIBIO, CEUA00086741; A. Ricarte det. 2006 (GBIF) as *Orthonevra onytes* (Séguy, 1961) • 1 ♂; same data as for preceding; malaise trap maR2, 9317, Syrphidae, genitalia missing, CEUA-CIBIO, CEUA00086735; A. Ricarte det. 2006 as *Orthonevra brevicornis* (Loew, 1843) • 1 ♂; same locality as for preceding; 14 April 2004; A. Ricarte leg.; sampling point Pa2, 149, *Pyrus* sp. (underneath the label), Syrphidae; CEUA-CIBIO, CEUA00086733; A. Ricarte det. 2006 (GBIF) as *Orthonevra brevicornis* (Loew, 1843) • 1 ♂; Salamanca, Campanarios de Azaba, Salamanca; 12 April 2011; Quinto, García, Quirce leg.; Malaise 2, Syrphidae; CEUA-CIBIO, CEUA00107961; A. Ricarte det. 2015 as *Orthonevra brevicornis* (Loew, 1843) • 1 ♂; same data as for the holotype; A. Ricarte leg; DNA CEUA_S60; CEUA-CIBIO, CEUA00107962 • 1 ♀; Ciudad Real, P. N. Cabañeros; 18 March–12 April 2005; A. Ricarte leg.; malaise trap maR1, 6895, Syrphidae; CEUA-CIBIO, CEUA00086742; A. Ricarte det. 2006 (GBIF) as *Orthonevra onytes* (Séguy, 1961) in Ricarte [34] • 1 ♀; same data as for preceding; 7507, headless; CEUA-CIBIO, CEUA00086740; A. Ricarte det. 2006 (GBIF) as *Orthonevra onytes* (Séguy, 1961) • 1 ♀; Jaén, La Iruela, Arroyo del Valle meadow, next to A-319 road; 37°55′12.54″ N, 2°56′57.37″ W; 1006 m; 1 June 2018; A. Ricarte leg.; with flowering *Crataegus* sp.; INV09450; A. Ricarte det. as *Orthonevra* sp. • 1 ♀; same data as for the holotype; A. Ricarte leg.; DNA CEUA_S61; CEUA-CIBIO, CEUA00107977 • 1 ♀; same locality as for the holotype; 27 February 2021; A. Ricarte and Z. Nedeljković leg.; DNA CEUA_S59; CEUA-CIBIO, CEUA00107960 • 1 ♀; same data as for preceding; CEUA-CIBIO, CEUA00107979 • 2 ♀♀; same data as for the holotype; A. Ricarte leg.; CEUA-CIBIO, CEUA00107975, CEUA00107978 • 1 ♀; same data as for the holotype; CEUA-CIBIO, CEUA00107976 • 1 ♀; same locality as for the holotype; 27 February 2021; A. Ricarte and Z. Nedeljković leg.; CEUA-CIBIO, CEUA00107974. The holotype and most male paratypes have the genitalia dissected and stored in a plastic microvial.

*Examined material of similar species*. *Orthonevra incisa*. POLAND • 2 ♂♂, 2 ♀♀; Biebrzański PN, Grobla Honczarowska “4”, p. Mal. (gora), na 4 “kw”; 23 May 2006; J. Sawoniewicz leg.; prep. RŻ’19D, zar. *Salix*; CEUA-CIBIO, CEUA00110206 to 00110209; R. Żóralski det. • 1 ♀; same data as for preceding (without indication of plant); 20 May 2008; CEUA-CIBIO, CEUA0010210.

*Diagnosis*. This species can be distinguished from all other *Orthonevra* species by the following combination of features: basoflagellomere 1–1.2× longer than wide, in females usually shorter than in males (Figure 5B,D); katepisternum with two separate patches of hairs and, in between, a patch of shorter hairs; metafemur with black hairs ventrally; sternum I pollinose; male genitalia as in Figure 7, with surstylus tapering towards the apex in lateral view, and hypandrium with asymmetric superior lobes bearing a spur on their inner side; in females, tergite V folds laterally giving it a triangular shape in dorsal view (Figure 6C), usually bearing a small apical keel (Figure 6B) that, in dorsal view, looks like a tooth or spina (Figure 6C); posterior margin of tergum V with a central incision reaching the keel.

MALE *(holotype)*. *Overall appearance* (Figure 4A,B). Black body with dark olive-green reflections; L = 6.3 mm (5.2–6.3 mm, *n* = 4); WL = 5.3 mm (4.3-5.3 mm, *n* = 4). *Head* (Figure 5A). Eye with very short—but obvious—scatted hairs; scape black; pedicel black, turning lighter at the apex, with a few long white hairs ventrally, and some shorter black hairs dorsally; basoflagellomere orange, blackish brown dorso-apically (Figure 5B); arista not longer than antenna, blackish brown, virtually bare (Figure 5B); basoflagellomere 1.1× longer than wide (1.1–1.2×, *n* = 4) (Figure 5B); head hairs all white to light yellow (Figure 5A); ocellar triangle slightly elevated, virtually equilateral; vertical triangle with long forwardly curved whitish-yellow hairs; eye contiguity as long as the ocellar triangle; frontal triangle coarsely punctured, shiny, with sparse bronze pollinosity, slightly excavated above lunule; frontal triangle with whitish-yellow hairs, shorter ventrally; lunule shiny black; face with a fascia of white pollinosity just below antennal insertions, from eye to eye; face finely wrinkled; in profile, face straight below antennae, then curving forwards until the projected ventral part (Figure 5A); occiput pollinose; in profile, ventral part of head straight (Figure 5A). *Thorax*. Black; wing wholly microtrichose, with veins black, except for basal part of Sc and R1; pterostigma brownish black; vein M_1_ recessive (Figure 6A); squama yellowish white; halter light orange; legs black (Figure 4A); leg hairs white to light yellow, except for the black hairs on the ventral part of metafemur and the setulae on tarsi; leg hairs longer on the postero-ventral part of pro- and mesofemora; profemur with an antero-ventral semi-circular lamina apically; mesonotum with short white to light yellow hairs, somewhat longer and slightly backwards inclined on the anterior half of scutum, and also longer (but straight) on the posterior margin of scutellum; scutellum with a slight sulcus posteriorly; pleuron with white to light yellow hairs, except for the bare anterior anepisternum (ventrally with a small patch of hairs), dorso-medial and posterior anepimeron, katepimeron, meron and metasternum; katepisternum with a dorsal patch of hairs and a ventral patch, as well as a third patch of shorter hairs in between; anterior spiracle light orange; posterior spiracle light yellow. *Abdomen*. Oval, black; in dorsal view, slightly wider than scutum at the maximum scutum width (Figure 4B); thickly pollinose (dull) except for shiny lateral margins of all terga; with short white to light yellow hairs, conspicuously longer on the anterior corners of tergum II and very short on the central areas of terga; all sterna shiny except for the entirely pollinose sternum I; sterna black (Figure 4A), with long erect hairs on sternum II, shorter and slightly inclined hairs on sterna I and III, hairs on sternum IV recumbent; genitalia as in Figure 7, with surstylus tapering towards the apex in lateral view, and hypandrium with asymmetric superior lobes bearing a spur on their inner side; apico-dorsal lobe of aedeagus with a developed hook-like process pointing towards the outer side of the hypandrium, and a smaller hook-like process basal to the large one.

FEMALE. Same as male except for the following: L = 6.4–7 mm (*n* = 5); WL = 5.2–6 mm (*n* = 5); basoflagellomere 1–1.2× (*n* = 6) longer than wide (Figure 5D); frons with hairs somewhat longer than scape, longer and forwardly inclined on the ocellar triangle; frons transversally wrinkled (Figure 4D), except for a medial smooth line. Anterior and posterior spiracles grey. Abdomen black, roundish oval, in dorsal view, wider than scutum at the maximum scutum width (more than in male) (Figure 4D); tergum V triangular in dorsal view, with each lateral margin of triangle twice, or more, the medial length of the triangle (Figure 6C); lateral sides of tergum V might or might not be visible in dorsal view, but always inclined downwards (Figure 6B); posterior corner of the triangular part of tergum V usually bearing a small keel (Figure 6B) that, in dorsal view, is seen as a tooth or spina (Figure 6C); tergum V with a central incision reaching the keel; sterna black (Figure 4C), somewhat lighter on sterna I and II; sternum I with erect hairs; sternum II with semi-recumbent hairs; sterna III to V with recumbent hairs.

*Etymology*. The specific epithet ‘arcana’, formed from the Latin ‘arcanum’ for secret or mystery, refers to the fact that the status of this species has proved difficult to resolve, as a mystery.

*Range*. Spain (provinces of Ciudad Real, Jaén, Madrid and Salamanca).

*Biology*. Adults visit flowers of *Salix atrocinerea* Brot. near small rivers where the larvae of this species are suspected to find their microhabitats. The species has also been recorded visiting flowers of wild *Pyrus* sp. (*Pyrus bourgaeana* Decne.) and *Crataegus* sp. Most records are from late winter / early spring (February to April), except for one that was collected as late as early June.

*Taxonomic notes*. This species is similar to *O. incisa* in having the sternum I entirely pollinose, and in females, a conspicuous incision on the posterior margin of tergum V. However, the new species can be separated from *O. incisa* by the shape of vein M_1_, which is recessive in the new species (as in all *Orthonevra*) (Figure 6A) but not recessive in *O. incisa* [7] (figure 12C). Males can also be clearly separated by the shape of their genitalia (Figure 7 and Figure 14). In the female, the triangle of the tergum V seen dorsally is conspicuously shorter in *O. arcana* sp. n. (Figure 6C) than in *O. incisa* (Figure 13C). The new species can be readily separated from *O. plumbago* by the pollinose sternum I, shiny in *O. plumbago*.

### 3.2. Chrysogaster coerulea *Strobl* in *Czerny and Strobl, 1909* stat. n.

The species *Chrysogaster coerulea* is represented in the Figure 8, Figure 9, and Figure 10C.

Gabriel Strobl, in Czerny and Strobl [35], proposed *coerulea* as a new variety of *Chrysogaster chalybeata* Meigen. Essentially, *coerulea* was used to name the Spanish specimens with a bluish head, thorax and abdomen (specially females) and with the central part of the wing more darkened than in the typical form (both sexes). Strobl [35] wrote on page 207 ‘*chalybeata* Mg., *basalis* Str. (non Loew?), var. *coerulea* m. Escorial, 2 ♂♂, 4 ♀♀, Spanien, ♂, ♀ (L)’, i.e., he was apparently referring to eight specimens collected by Jorge Lauffer (‘L’), six from ‘Escorial’ and two from ‘Spanien’. Strobl [35] suggested that *C. basalis* of Loew could in fact be a variety of *C. chalybeata* based on the observed similarity with the new *coerulea*. Whether all eight mentioned specimens belonged to the new variety or also included specimens of *C. basalis* examined by Strobl is uncertain.

In the Strobl’s collection, which is deposited in Admont, Austria (BSA), there are not specimens labelled as *coerulea*, but the variety is listed on page 689 of the main BSA handwritten Collection Catalogue as ‘v. *coerulea* m. Spanien I ♂ II ♀’ [36]. Evenhuis and Pape [37] assume that the type series for *coerulea* of Strobl consists of two males and four females, but this is uncertain under the light of the BSA Collection Catalogue, which refers to one male and two females. In the MNCN collection, there are several specimens of Lauffer catalogued as ‘*Chrysogaster* sp.’, but none of those have identification labels. However, other non-*Chrysogaster* specimens from the MNCN bear Strobl’s identification labels. So, we interpret that the Lauffer’s unlabelled specimens of *Chrysogaster* from the MNCN do not belong to the type series of *coerulea*, even if seven of these unlabelled specimens are from ‘El Escorial’ and could have been collected at the same time as the type material of *coerulea*. In fact, Gil-Collado [16] states that there are no specimens of *C. chalybeata*, neither of the typical form, nor of *coerulea*, in the MNCN collection at the time he studied it.

Evenhuis and Pape [37] consider that the name *coerulea* should be treated as infrasubspecific and then unavailable. The valid name Evenhuis and Pape [37] give for *coerulea* is *Chrysogaster coemiteriorum*, as *C. chalybeata* is regarded as a junior synonym of *C. coemiteriorum*. However, there are not enough grounds to believe that *coerulea* was described as an infrasubspecific entity. Strobl [35] described a geographical subspecies when proposing *coerulea* for specimens from a certain region, Spain, where the typical (‘European’) form of *C. chalybeata* was meant to be absent by then. Thus, following the article 45.6.4 of the Code, the name *coerulea* is to be treated as subspecific.

The seven *Chrysogaster* specimens from ‘El Escorial’ of Lauffer’s collection (MNCN) were examined, and the males shared genitalia morphology with recently collected males from other Spanish localities. One male from Lauffer’s collection did not have a blue shine, probably because it was a teneral when collected. However, the males from other Spanish localities and specially their females had a conspicuous blue shine, as suggested for *coerulea* in Strobl’s description. We interpret all these specimens correspond to Strobl’s concept of *coerulea*, and they clearly differ from *C. coemiteriorum* and *C. basalis* in various features including the male genitalia, which is different from any other genitalia of *Chrysogaster* species in the literature. Thus, the status of *coerulea* is now formally upgraded to species level as *Chrysogaster coerulea*, a name already used in Peck’s catalogue [5] as a synonym of *C. chalybeata* but without further explanation.

The species concepts of *C. coerulea* and allies have never been clear. For example, Gil-Collado [16] illustrated the head of *C. basalis* in lateral view, but his drawing had in fact a more protruding facial tubercle than in the specimens currently studied of this species. The present paper shows that *C. coerulea* has a more protruding facial tubercle than any other similar species, and Gil-Collado [16] possibly drew the head of *C. coerulea*. This evident confusion together with the fact that the type series of *C. coerulea* is lost suggest the need to clarify the taxonomic status of *C. coerulea* through the designation of a neotype and detailed redescription of this taxon. The specimen designated here as neotype is a male from the original type locality, El Escorial, and by the same collector as the original type material, Lauffer (MNCN). The male, which has both slight blue shine and wing pigmentation (teneral?), shares genitalia morphology with recently collected males showing heavier blue shine and wing pigmentation. Upon the examination of all available material, the darker wing referred by Strobl for *C. coerulea* appears to be somewhat variable and so, not a good character as the facial tubercle or male genitalia.

*Type material*. NEOTYPE. SPAIN • ♂; Escorial; Lauffer leg.; colección Lauffer; MNCN, MNCN_Ent 301518 (blue label); as *Chrysogaster basalis* in Gil-Collado [16].

*Additional material*. ***New.*** SPAIN • 4 ♂♂, 1 ♀; Segovia, La Granja, arroyo del Pto. del Paular, 40°49′24.6″ N, 3°57′49.1″ O; 1825 m; 28 July 2021; A. Ricarte et al. leg.; DNA CEUA_S107; CEUA-CIBIO, CEUA00109499, CEUA00109502, CEUA00111500 • 1 ♂; Madrid, Cercedilla, borde Ctra. M-622; 40°44′10″ N, 4°2′15.7″ O; 1200 m; 28 July 2021; I. Ballester leg.; CEUA-CIBIO, CEUA00109503 • 1 ♂, 1 ♀; Granada, Sierra Nevada, Monachil, Museo Fuente Alta; 2125 m; 22 June 2021; A. Ricarte leg.; CEUA-CIBIO, CEUA00109504, CEUA00111513 • 2 ♂♂; Granada, Sierra Nevada, Monachil, Pradollano, Estación de esquí; 2180 m; 20 June 2021; P. Aguado Aranda leg.; DNA CEUA_S108; CEUA-CIBIO, CEUA00109505 to 00109506 • 1 ♀; Granada, Sierra Nevada, Güejar Sierra, El Dornajo, Parking camino a Peña del Perro; 1895 m; 26 June 2021; I. Ballester Torres; DNA CEUA_S109; CEUA-CIBIO, CEUA00111539 • 1 ♂; Teruel, Orihuela del Tremedal; 8 July 1997; M. A. Marcos García leg.; CEUA-CIBIO, CEUA00109508 • 1 ♂, 2 ♀♀; Soria, Covaleda de la Sierra; 1200 m; 21 July 1989; M. A. Marcos García leg.; CEUA-CIBIO, CEUA00109507, CEUA00109509 to 00109510 • 1 ♀; same data as for preceding; CEUA-CIBIO, CEUA00111510 • 3 ♀♀; Madrid, Rascafría, arroyo de la Angostura; 40°50′4″ N, 3°54′42″ O; 1430 m; 27 July 2021; A. Ricarte leg.; CEUA-CIBIO, CEUA00111501 to 00111503 • 1 ♀; Palencia, Sierra María de Redondo, Cueva del Cobre; 1500 m; 22 August 1995; M. A. Marcos García leg.; CEUA-CIBIO, CEUA00111512 • 1 ♀; Cañamero, Rio Ruelos, Cáceres; 8 June 1980; M. A. Marcos García leg.; CEUA-CIBIO, CEUA00001659. All unpublished males have the genitalia dissected and stored in a plastic microvial. ***Revised***. SPAIN • 1 ♀; León, Crémenes; 16 July 1987; M. A. Marcos García leg.; CEUA-CIBIO, CEUA00017076; as *C. cemiteriorum* in Marcos-García [38] • 1 ♀; Madrid; Dusmet leg.; MNCN, MNCN_Ent 301418; as *C. macquarti* in Gil-Collado [16] • 1 ♀; Ávila, Navarredonda; June 1909; Exp. del Museo; MNCN, MNCN_Ent 301419; as *C. macquarti* in Gil-Collado [16] • 1 ♂; Cercedilla; J. Arias leg.; MNCN, MNCN_Ent 301416; as *C. basalis* in Gil-Collado [16] • 1 ♀; Ciudad Real, P.N. Cabañeros; 10 June 2006; A. Ricarte leg.; sampling point Mm1; CEUA-CIBIO, CEUA00082499; as *Chrysogaster basalis* in Ricarte [34] • 2 ♂♂; Salamanca, Pto. Vallejera; 1202 m; 24 August 1980; M. A. Marcos García leg.; CEUA-CIBIO, CEUA00017057; as *C. basalis* in Marcos-García [39] • 7 ♂♂; Linares de Riofrío; 13 June 1979; M. A. Marcos- García leg.; CEUA-CIBIO, to 00017063; as *C. basalis* in Marcos-García [39] • 1 ♂; Salamanca, Rinconada de la Sierra; 11 July 1980; M. A. Marcos- García leg.; CEUA-CIBIO, CEUA00017056; as *C. basalis* in Marcos-García [39] • 1 ♀; Cabezabellosa; 25 August 1980; M. A. Marcos García leg.; CEUA-CIBIO, CEUA00001660; as *C. basalis* in Marcos-García [39] • 1 ♀; Salamanca, Caminomorisco; 2 July 1980; M. A. Marcos García leg.; CEUA-CIBIO, CEUA00001666; as *C. basalis* in Marcos-García [39] • 2 ♀♀; Eljas; 3 July 1980; M. A. Marcos García leg.; CEUA-CIBIO, CEUA00017070, CEUA00017072; as *C. basalis* in Marcos-García [39] • 2 ♀♀; Cáceres, Gata; 3 July 1980; M. A. Marcos García leg.; CEUA-CIBIO, CEUA00001661, CEUA00001663; as *C. basalis* in Marcos-García [39] • 1 ♂; Cádiz, Pto Galis (subida); 13 June 1984; M. A. Marcos García leg.; CEUA-CIBIO, CEUA00007885 • 3 ♀♀; same locality as for preceding; 12 June 1984; M. A. Marcos García leg.; CEUA-CIBIO, CEUA00001668 to 00001670; as *C. basalis* in Marcos-García [40] • 5 ♀♀; Ávila, Becedas; 1200 m; 24 August 1980; M. A. Marcos García leg.; CEUA-CIBIO, CEUA00001665, CEUA00017066 to 00017067, CEUA00017069, CEUA00017071; as *C. basalis* in Marcos-García [39] • 1 ♀; Cáceres, Tornavacas, 30 Sepeptember 1980, M. A. Marcos García leg.; CEUA-CIBIO, CEUA00001662; as *C. basalis* in Marcos-García [39] • 1 ♀; Salamanca, Montemayor del Río; 23 September. 1980; M. A. Marcos García leg.; CEUA-CIBIO, CEUA00017068; as *C. basalis* in Marcos-García [39] • 1 ♀; Cáceres, Descargamaría; 2 July 1980; M. A. Marcos García leg.; CEUA-CIBIO, CEUA00017073, as *C. basalis* in Marcos-García [39] • 1 ♀; Santibáñez de la Sierra, Puente Alagón; 24 September 1980; M. A. Marcos García leg.; CEUA-CIBIO, CEUA00017065; as *C. basalis* in Marcos-García [39] • 1 ♀; Salamanca, Pto Perales; 850 m; 16 August 1980; M. A. Marcos García leg.; CEUA-CIBIO, CEUA00001664; as *C. basalis* in Marcos-García [39] • 1 ♀; Ciudad Real, P. N. de Cabañeros; 6 July 2005; A. Ricarte leg.; Sampling point Pa2; CEUA-CIBIO, CEUA00082501; as *C. basalis* in Ricarte [34] • 2 ♀♀; same locality as for preceding; Ciudad Real, P.N. de Cabañeros; 10 June 2006; A. Ricarte leg.; sampling point Mm1; CEUA-CIBIO, CEUA00082500; as *C. basalis* in Ricarte [34] • 1 ♀; Ciudad Real, P.N. de Cabañeros; 27 June 2006; A. Ricarte leg.; sampling point xMa2; CEUA-CIBIO, CEUA00082502; as *C. basalis* in Ricarte [34] • 7 ♀♀; Escorial, Lauffer leg.; MNCN, MNCN_Ent 301509, MNCN_Ent 301511, MNCN_Ent 301513 to 301517; as *C. basalis* in Gil-Collado [16] • 1 ♀; Cercedilla; Arias leg.; MNCN, MNCN_Ent 301409; as *C. chalybeata* in Gil-Collado [16] • 2 ♀♀; Sierra de Guadarrama; Dusmet leg.; MNCN, MNCN_Ent 301469 to 301470; as *C. chalybeata* in Gil-Collado [16]; A. Ricarte 2021 det. as *C. virescens* • 1 ♀; without data; collection Andréu; UMCZ.

*Redescription*. MALE (*based on a specimen with typical colouration of this species from ‘La Granja, Segovia’*). *Overall appearance* (Figure 8). Black body with bluish reflections; L = 7.03 mm (6.19–7.31 mm, *n* = 4); WL = 5.1 mm (4.6–5.75 mm, *n* = 4). *Head* (Figure 10C). Eye with short sparse yellow hairs, almost bare in the ventral part; scape black; pedicel brownish with a few hairs ventrally and dorsally; basoflagellomere brownish, orange-reddish ventrally (Figure 10C); arista longer than antenna, dark brown, bare; basoflagellomere roundish (Figure 10C), 1.25× longer than wide (0.95–1.5×, *n* = 4); ocellar triangle conspicuously elevated, equilateral; vertical triangle with long black hairs; eye contiguity two times longer than the vertical triangle, frontal triangle shiny black, with medial excavation from antennal insertions to posterior 2/3 of length of frontal triangle; frontal triangle with long black hairs, lunule shiny, black; face with a fascia of brown sparse pollinosity just below antennal insertions, from eye to eye, the fascia about as wide as a third of the face width; face finely wrinkled laterally, with a slight but conspicuous shiny black tubercle medially; face with short, scarce yellow hairs, except for the bare area below antennal insertions and tubercle; occiput with very sparse pollinosity, much denser on the eye margin; occiput with black hairs. *Thorax*. Wing wholly microtrichose, with veins black to dark brown, except for orange basal part of Sc, CuA, CuP and A1 (Figure 8A,C); pterostigma brownish black; wing with a conspicuous large brown pigmented area antero-apically (Figure 8A,C); hairs on vein C black; calypteres yellowish white; halter with pedicel orange, but blackish base and capitulum; legs black; leg hairs black, except yellow hairs in the anterior part of meso- and metafemora; meso- and metafemora with two rows of black setulae ventrally; setulae on pro- and metatarsi yellow, on mesotarsi black; legs sparsely pollinose except profemora which are shiny posteriorly; scutum shiny, black with bluish reflections, sparsely (and inconspicuously) pollinose anteriorly; scutum with semi-recumbent black hairs and black setulae intermixed; scutellum shining black with long black hairs in the posterior margin; scutellum with a posterior slight sulcus; pleuron with white to light yellow hairs, except for the bare anterior anepisternum (ventrally with a small patch of hairs), posterior anepimeron, katepimeron, meron and metasternum; katepisternum with a large bare area between the dorsal and ventral patches of hairs; anterior spiracle light orange; posterior spiracle dark brown. *Abdomen*. Oval-shaped (Figure 8B); thickly pollinose (dull) except for shiny lateral margins of terga II–IV, and posterior margin of tergum IV; terga with light yellow short and recumbent hairs; tergum I with long yellow hairs on the lateral margins, tergum II with long yellow recumbent hairs in its anterior part; all sterna shiny except for the entirely pollinose sternum I; sterna with white hairs, erect on sternum I, shorter and recumbent on sterna II–IV; genitalia as in Figure 9, with the superior lobes of the aedeagus cover elongated, straight and approximated to each other along most of their length (Figure 9B,C); aedeagus cover with blunt upper corners (‘shoulders’) in ventral view (Figure 9C).

FEMALE (Figure 8B,D). Same as male except for the following: L = 5.96–7.84 mm; WL = 4.48–5.86 mm; basoflagellomere 0.9–1.03× longer than wide (*n* = 4). L = 6.35 mm; WL = 5.31 mm (*n* = 4); usually with more vivid blue reflections than males; basoflagellomere 0.99 × longer than wide (*n* = 4), brown; frons with scarce short, yellow hairs, as long as diameter of ocellus or nearly so; frons surface angled near antennae; posterior part of frons with a differentiated medial sulcus of conspicuous width, sulcus with lower hair density; anterior part of frons (posterior to the frons angle) with lateral wrinkles diagonal to eye margin; face flat, without tubercle; pollinose fascia of face of triangular shape; mesonotum only with short black setulae; abdomen roundish oval, slightly wider than scutum at the maximum scutum width (Figure 8D); tergum V entirely shiny; hairs of terga following the same pattern as in male; sterna with shorter hairs than in male, all semi-recumbent.

*Range*. Spain (provinces of Ciudad Real, Granada, Huesca, León, Madrid, Salamanca, Soria and Teruel).

*Biology*. Adults visit flowers of *Euphorbia nicaeensis* All. (Euphorbiaceae) and *Magydaris panacifolia* (Apiaceae). They can be found in localities near streams (e.g., in ‘La Granja, arroyo del Pto. del Paular’) where their larvae are suspected to find their breeding sites. Adult records are from mid-June to late August.

*Taxonomic notes.* Males of this species can be distinguished from the similar *C. basalis* by the presence of a facial tubercle, which is absent in *C. basalis* (Figure 10A). The facial tubercle of *C. coerulea* is more expressed than that in *C. solstitialis* (Figure 10B). From the similar *C. solstitialis*, *C. coerulea* can be separated by the shiny or just partly pollinose proepimeron, which is extensively pollinose in *C. solstitialis*. The wing base of *C. coerulea* is light yellow, whilst in *C. solstitialis* is dark brown. The females of *C. coerulea* can be distinguished from those of other *Chrysogaster* species by the combination of blue reflections in the thorax and orangey yellow wing base (Figure 8B,D). However, females of *C. coerulea* and *C. basalis* are sometimes difficult to separate from each other, because there are some females of *C. coerulea* with less conspicuous bluish shine and some females of *C. basalis* with blue shine at least in the pleuron.

The facial tubercle of *C. coerulea* is also more expressed than in *C. mediterraneus*, in which it is just slightly produced [9] (figure 1). Male of *C. coerulea* is also similar to *Chrysogaster musatovi* Stackelberg, 1952 in the straight and approximated superior lobes of the aedeagus cover, but *C. musatovi* differs, for example, (a) in the facial tubercle, which is inconspicuous in *C. musatovi* (see photo of the holotype at http://www.zin.ru/collections/Diptera/specimen_en.html?Catalog_UID=1345139328343005 accessed on 1 February 2022) but obvious in *C. coerulea* (Figure 10C), and (b) in the shape of the surstylus [41] (figure 5) (Figure 9A).

The male genitalia of *C. coerulea* (Figure 9) are similar to those of *C. basalis*, *C. solstitialis* and *Chrysogaster mediterraneus* Vujić, 1999. Unlike the other *Chrysogaster* species, the superior lobes of the aedeagus cover of *C. coerulea* are straight and approximated to each other along most of their length (Figure 9B,C). In the ventral view (Figure 9C), the aedeagus cover of *C. coerulea* has much less pronounced ‘shoulders’ than in *C. solstitialis*. The superior lobes of the aedeagus cover of *C. mediterraneus* are shorter than in *C. coeruela*. 

### 3.3. The Holotype of *Orthonevra incisa* (Loew, 1843)

*Examined material*. Hermann Loew described this species from a female collected in ‘Posen’ on 1st May (1843). Maibach et al. [7] indicated that there is a female in Loew’s collection (ZMB, Berlin) that is, unquestionably, the type of *C. incisa*. This specimen was located in the ZMB by the authors of the present paper and confirmed to be the type of this taxon. The specimen, in good condition, is labelled as follows: 1♀, ‘Posen 1.5.43’ (handwritten), ‘Coll. H. Loew’ (printed), ‘12921’ (printed), ‘Type’, ‘*Chrysogas. incisa* m.’, ‘Zool. Mus. Berlin’ (printed on yellow label), ‘Zool. Mus. Berlin’ (printed on white label), ‘HOLOTYPUS *Chrysogaster incisa* Loew, 1843 Des. A. Maibach 1992’ (Figure 11C).

*Redescription*. FEMALE HOLOTYPE. *Overall appearance* (Figure 11A,B). Black body with bluish grey reflections; L = 6.7 mm; WL = 6.1 mm. *Head* (Figure 12A). Eye bare, with scattered hairs only visible at high magnification; scape blackish orange; pedicel blackish orange, with a few long yellowish white hairs ventrally, and some shorter hairs of the same colour dorsally; basoflagellomere orange, slightly darker dorso-apically; arista not longer than antenna, dark orange, virtually bare; basoflagellomere roundish (0.9× longer than wide) (Figure 12B); head hairs all white to light yellow; ocellar triangle slightly elevated; distance between hind ocelli slightly greater than distance between the front ocellus and a hind ocellus; frons with short hairs (as long as scape), somewhat longer and forwardly inclined on the ocellar triangle; frons transversally wrinkled on eye margins, smooth along a medial line; lunule orangey brown; face with a fascia of white pollinosity just below antennal insertions, from eye to eye; face slightly wrinkled, smoother on the forwardly projected ventral part; in profile, face straight for a short length below antennae, then curving forwards until the projected ventral part; occiput pollinose, more sparsely towards the vertex; in profile, ventral part of head gently curved towards the projected part. *Thorax*. Black; wing wholly microtrichose, with veins brown to orange; pterostigma light orange; vein M_1_ non-recessive, with its anterior end almost perpendicular to R_4+5_ (Figure 12C); squama yellowish white; halter orange; legs dark brown to black; leg hairs white to light yellow, except for some black setulae on the ventral part of tarsi; leg hairs longer on the postero-ventral part of pro- and mesofemora; profemur with an antero-ventral semi-circular lamina apically; mesonotum with short white to light yellow hairs, both slightly longer and backwards inclined on the anterior half of scutum, and also longer (but straight) on the posterior margin of scutellum; scutellum with a slight sulcus posteriorly; pleuron with short white to light yellow hairs, except for the bare anterior anepisternum (ventrally with a small patch of hairs), dorso-medial and posterior anepimeron, katepimeron, meron and metasternum; katepisternum with a dorsal and a ventral patch of hairs; anterior spiracle light grey; posterior spiracle orange. *Abdomen*. Oval, brownish black; in dorsal view, wider than scutum at maximum scutum width; sparsely pollinose (dull) except for shiny lateral margins of all terga; with short white to light yellow hairs, conspicuously longer on the anterior corners of tergum II; tergum V with slightly longer hairs than the other terga (Figure 13A); tergum V triangular in dorsal view, lateral margin of triangle less than twice the medial length of the triangle (Figure 13C); lateral sides of tergum V folded giving the triangular dorsal appearance (Figure 13A,B); posterior margin of tergum V with a central incision nearly reaching the tip of the dorsal triangular part of the tergum (Figure 13B); all sterna shiny except for the entirely pollinose sternum I; sterna orange, darker in sternum I and V, with white short recumbent hairs.

*Taxonomic notes*. During the course of the present study, five specimens of *O. incisa* from Poland were examined and compared with the female holotype, also from Poland. The Polish females were conspecific with the female holotype but somewhat variable in the shine colour of body (darker in the recent females), scape colour (black in the recent females), colour of wing veins (black to blackish brown, except for the orange basal sections of veins, in the recent females), and colour of metafemur hairs, which are all white in the holotype (Figure 12D) (some black scattered hairs might be present ventrally, on the apical half of metafemur in the recent females). Variation is attributed to the age of the holotype specimen (179 years old) or to the fact that this was more teneral when collected in 1843. The Polish males (Figure 14), also considered conspecific with the female holotype, had the same genitalia as that illustrated for *O. incisa* by Stackelberg [42].

### 3.4. The Lectotype of *Orthonevra plumbago* (Loew, 1840)

*Examined material*. Hermann Loew described this species from the ‘area of Posen’, Poland from an undetermined number of specimens of unknown sex. According to Evenhuis and Pape [37], the type of *O. plumbago* is deposited in the ZMB collection (Berlin), where we found two—apparently—conspecific females labelled as ‘typus’: 1♀, ‘plumbago Lw’ (handwritten), ‘Coll. H. Loew’ (printed); 1♀, ‘Orthoneura plumbago m’ (handwritten), ‘12929’ (printed), ‘Coll. H. Loew’ (printed), ‘Zool. Mus. Berlin’ (printed), ‘96’ (handwritten in a triangular label, doubtful writing) (Figure 15C). The first female is abdomenless and wingless, while the second is in good condition. Hermann Loew used ‘m’ after a name to indicate a taxon was described by him; this pattern is also found in the labels of the holotype of *Chrysogaster incisa* Loew, 1843 (see under this species section). Given that the two females found are also labelled as types, and they meet (at least the complete female) the features provided in the original description of *O. plumbago*, we consider them part of the type series and designate here the female labelled ‘Orthoneura plumbago m’ as lectotype (the other becomes a paralectotype).

*Description*. FEMALE LECTOTYPE. *Overall appearance* (Figure 15A,B). Black body with bluish reflections; L = 6.3 mm; WL = 5.4 mm. *Head* (Figure 16A). Eye bare; scape blackish brown; pedicel brown, with a few long white hairs ventrally, and some shorter white and brown hairs intermixed dorsally; basoflagellomere brown, baso-ventrally orange; arista not longer than antenna, brown, virtually bare; basoflagellomere elongate (1.3× longer than wide), with round apex (Figure 16B); head hairs all white to light yellow; ocellar triangle slightly elevated; distance between hind ocelli greater than distance between the front ocellus and a hind ocellus; frons with short hairs (as long as scape), longer and forwardly inclined on the ocellar triangle; frons transversally wrinkled on eye margins, smooth along a medial line that tapers towards the ocellar triangle; lunule brown, contrasting with the darker frons; face with a narrow fascia of white pollinosity just below antennal insertions, from eye to eye; face slightly wrinkled, smooth on the forwardly projected ventral part; in profile, face just slightly convex, almost straight; occiput sparsely pollinose, densely along a narrow line on eye margin; in profile, ventral part of head entirely straight. *Thorax*. Black; wing wholly microtrichose, with veins brown to orange (Figure 16C); pterostigma yellowish orange; vein M_1_ recessive (Figure 16C); squama whitish orange; halter orange; legs dark brown to black, with somewhat lighter coxae and trochanters; leg hairs white to light yellow, except for some black spiny hairs on the ventral part of the apical section of metafemur, and some black setulae on the ventral part of tarsi; leg hairs longer on the postero-ventral part of pro- and mesofemora; mesonotum with short (not longer than scape), erect, white to light yellow hairs; scutum with 3–4 inconspicuous vittae on the anterior margin; scutellum with a slight sulcus posteriorly; pleuron with short white to light yellow hairs, except for the bare anterior anepisternum (ventrally with a small patch of hairs), dorso-medial and posterior anepimeron, katepimeron, meron and metasternum; anterior spiracle grey; posterior spiracle brown. *Abdomen*. Oval, brownish black; in dorsal view, wider than scutum at the maximum scutum width; sparsely pollinose (dull) except for shiny lateral margins of all terga; with short white to light yellow hairs, conspicuously longer on the anterior corners of tergum II, on the posterior margin of tergum IV, and on tergum V; tergum IV with a small wart-like elevation in the middle of posterior margin; posterior corners of tergum IV downwards directed, forming a triangle continuous with tergum V in dorsal view; tergum V with a medial sulcus starting on the posterior margin as a narrow short incision (partly broken) and then extending almost to the anterior margin as a seam (in posterior view, the sulcus divides the tergum giving it the appearance of two valves) (Figure 17B); in lateral view, tergum V rounded, with the sulcus dorso-ventrally orientated (Figure 17A); all sterna shiny, orange (sternum V darker), with white recumbent hairs, longer along the medial line of sterna and on the entire sternum V (Figure 17C).

*Taxonomic notes*. The lectotype of *O. plumbago* would not key out with existing keys. For example, with van Veen [43], the lectotype would key out only to couplet 2, because the basoflagellomere is neither as long as wide, nor 1.5−2× longer than wide. Even if the length/width ratio of the basoflagellomere of the lectotype is interpreted as intraspecific variability of an elongate basoflagellomere, the lectotype would not key out to *O. plumbago* because it has not a deep incision at hind margin of tergite V (Figure 17B), as indicated in van Veen [43]. With Bartsch et al. [44], the lectotype would key out to the couplet 4 (*O. brevicornis* and *O. plumbago*), but *O. plumbago* would be unreachable because the basoflagellomere of the lectotype is less than 1.5−2× longer than wide (Figure 16B), and the hind margin of tergite V does not have a sharp-edged ridge, but a seam all along most of the medial tergite length (Figure 17B). Speight and Castella [45] use a ‘provisional’ concept of *O. plumbago* based on Stackelberg [42] and indicate that this species female can be separated from *O. brevicornis* by the deep incision on the posterior half of the tergite V that sort of divides the tergite in two halves. Again, an incision as such, drawn also in Haarto and Kerppola [46], is absent from the proposed lectotype (Figure 17B).

Stackelberg [47], which modifies Stackelberg [42] in part, provides in a key some features approaching the here-designated lectotype to his own concept of *O. plumbago*; for example, he indicates that the female of *O. plumbago* has a small elevation on the posterior margin of tergite IV, as in the lectotype. However, he also indicates that the hairs on the mesonotum are 2−3 times longer than the arista base, but the lectotype has longer hairs in this body part, and that the narrow incision of the tergite V almost reaches its anterior margin, but the lectotype has a very short incision that soon becomes a seam towards the anterior margin (Figure 17B). Any of the mentioned references address the apparently characteristic curved orientation of the sulcus of tergite V in the lectotype (Figure 17A). 

A female of *O. plumbago* from Vorkuta (Russia) collected by Gerard Pennards was examined from photos. This female shared morphology with the lectotype, including the shape of the tergum V. However, the incision of the tergum V was deeper than that in the lectotype. Thus, the length of the incision appears to be variable in *O. plumbago*. Given the confusion found in the literature with this species, females of *O. plumbago* identified in Europe and the European parts of Russia should be now revised by comparison with the featured lectotype and reported variability in certain features (e.g. length of the posterior incision in tergum V of female). Regarding males *sensu* Stackelberg [42], further morphological and molecular analyses should be undertaken to confirm whether those of *O. plumbago* are conspecific with the lectotype-like females.

### 3.5. Key to the Studied Brachyopini genera

The key presented below is adapted from Maibach et al. [6].
Wing: basal section of radial vein (Figure 12C) with some long hairs dorsally … 2
-Wing: basal section of radial vein (Figure 12C) bare, without hairs … 3Male holoptic and female dichoptic; ventral side of metafemur with black setulae, at least apically; face, from antennae to mouth edge, flat or slightly convex in lateral view; basoflagellomere elongate, 1.5-4× longer than width … ***Riponnensia*** (go to Section 3.5.5)
-Male and female dichoptic; metafemur without setulae; face, from antennae to mouth edge, concave in lateral view; basoflagellomere large, roundish to oval, up to 1.5× longer than width … ***Lejogaster*** (go to Section 3.5.2)Lateral margins of tergum I entirely shiny; vein M_1_ usually recessive (as in Figure 6A); sternum I usually shiny … ***Orthonevra*** (go to Section 3.5.4)
-Lateral margins of tergum I pollinose, at least on the anterior fourth; vein M_1_ non-recessive (as in Figure 1D and Figure 8A); sternum I pollinose … 4Basoflagellomere brownish black (Figure 2A); tergum II with short to very short hairs on its central area; in male, face sometimes slightly concave, with a conspicuous medial tubercle (Figure 2A,C) … ***Melanogaster*** (go to Section 3.5.3)
-Basoflagellomere orange to red (Figure 10); tergum II often with long recumbent hairs on its central area; in males, face straight, usually with a less-conspicuous medial tubercle (Figure 10A,B), sometimes with a bumped area at each side … ***Chrysogaster*** (go to Section 3.5.1)

#### 3.5.1. *Chrysogaster* Meigen, 1803

Below is a key to the Iberian species of *Chrysogaster*. In many cases, dissection of the male genitalia is crucial to confirm the species identification in this genus.
Proepimeron completely pollinose … ***C. coemiteriorum***
-Proepimeron with a large central shiny area, sometimes barely pollinose … 2Sternum 2 with long hairs, of about two-thirds the length of the scutum hairs … ***C. rondanii***
-Sternum 2 only with short recumbent hairs, occasionally with a few longer hairs but never as long as in *C. rondanii* … 3Narrow-faced species; in frontal view, the lowest part of frons (female) is narrower than the maximum width of an eye or equal in width; males with frontal triangle of 90° or less; scutum, in males, with long inclined hairs at least on the anterior half … 4
-Broad-faced species; in frontal view, the lowest part of frons (female) wider than the maximum width of an eye; males with frontal triangle of more than 90°; scutum, in male, with long nearly erect hairs at least on the anterior half … ***C. virescens***Mesonotum shiny black; male with a small facial tubercle (Figure 10B) … ***C. basalis*** (females of this species cannot always be separated from *C. coerulea*, but females of *C. coerulea* usually have more bluish reflections than *C. basalis*)
-Mesonotum with blue reflections (Figure 8); male with a conspicuous facial tubercle (Figure 10C) … ***C. coerulea***


***Chrysogaster basalis* Loew, 1857**


*Examined material*. ***New*****.** SPAIN • 2 ♂♂, 1 ♀; Teruel, Frías de Albarracín; 7 July 1997; M. A. Marcos García leg.; CEUA-CIBIO, CEUA00109511 to 00109512, CEUA00111511 • 1 ♂; Teruel, Pto. El Cubillo; 1150 m; 6 July 1997; M. A. Marcos García leg.; CEUA-CIBIO, CEUA00109520 • 2 ♂♂; Huesca, Siresa; 20 July 1983; M. A. Marcos García leg.; CEUA-CIBIO, CEUA00017077 to 00017078 • 1 ♂; Huesca, Valle de Ansó; 850 m; 28–30 June 1943; MNCN, MNCN_Ent 301495; Peris Torres 1945 det. as *Liogaster metallina*. ***Revised***. SPAIN • 1 ♀; Huesca; 1338 m; 25 August 1983; M. A. Marcos García leg.; CEUA-CIBIO, CEUA00017075; published in Marcos-García [48] • 1 ♀; Huesca; 25 August 1983; M. A. Marcos García leg.; CEUA-CIBIO, CEUA00001667; as *C. chalybeata* in Marcos-García [48].

*Additional published records*. PORTUGAL • Speight [49] (without examined material); van Eck [12,13] • SPAIN • van der Goot [50] (as *C. chalybeata*); Gil-Collado [16] (specimens from Setcases and Sierra de Moncayo, as *C. chalybeata*, and specimens from Cercedilla, as *C. chalybeata* var. *coerulea*); Carles-Tolrá [51].


***Chrysogaster coemiteriorum* (Linnaeus, 1758)**


*Examined material*. ***New***. SPAIN • 2 ♂♂, 1 ♀; Teruel, Frías de Albarracín; 7 July 1997; M. A. Marcos García leg.; CEUA-CIBIO, CEUA00109514 to 00109515, CEUA00111514 • 1 ♂; Teruel, Griegos; 8 July 1997; M. A. Marcos García leg.; CEUA-CIBIO, CEUA00109516.

*Additional published records*. Marcos-García [38] reports a female, as ‘*C. cemiteriorum*’, from León, but this was in fact a female of *C. coerulea*. There are no additional records from the Iberian Peninsula published as ‘*C. coemiteriorum*’ or ‘*C. cemiteriorum*’.

*Notes*. The specimens examined by Andréu [52] were not found in the UMCZ collection (Murcia). Gil-Collado [16] transcribed the information provided by Andréu [52] for *C. chalybeata* (a synonym of *C. coemiteriorum*) but stated that there were no specimens of this species deposited in the MNCN collection (Madrid). There is also confusion with *C. basalis* in the literature because several records of *C. chalybeata* were actually reidentified as *C. basalis*, as indicated in the present study and in Vujić [9].


***Chrysogaster coerulea* Strobl *in* Czerny and Strobl, 1909**


See under Section 3.2.


***Chrysogaster rondanii* Maibach and Goeldlin de Tiefenau, 1995**


*Examined material*. ***Revised***. SPAIN • 1 ♂; Ávila, Sierra Candelario, Lag. Trampal; 2200 m; 28 June 1980; M. A. Marcos García leg.; CEUA-CIBIO, CEUA00017064; as *C. basalis* in Marcos-García [39].

*Notes*. This is the first record of this species from the Iberian Peninsula.


***Chrysogaster solstitialis* (Fallén, 1817)**


*Examined material*. ***New.*** SPAIN • 3 ♂♂; Covadonga; July 1928; J. Dusmet leg.; MNCN, MNCN_Ent 301475 to 301476, MNCN_Ent 301479 • 1 ♂; Prov. De Navarra, Valle del Irati, Bosque Irati; 3 July 1947; without leg.; MNCN, MNCN_Ent 301521 • 1 ♂; León, Candín; 4 August 1987; M. A. Marcos García leg.; CEUA-CIBIO, CEUA00017085 • 1 ♂; Asturias, Playa Ñora; 25 August 2019; M. A. Marcos García leg.; CEUA-CIBIO, CEUA00109517) • 1 ♂, 1 ♀; Madrid, Sierra de Guadarrama, Puerto de Cotos; 14 July 2019; A. Ricarte leg.; DNA CEUA_S74, CEUA_S102; CEUA-CIBIO, CEUA00109518 to 00109519 • 3 ♂♂, 5 ♀♀; León, Posada de Valdeón, Prada, bordes ctra.; 43°8′44.7″ N, 4°54′35.4″ O; 986 m; 25 August 2021; A. Ricarte and Z. Nedeljković leg.; CEUA-CIBIO, CEUA00109506 to 00109508, CEUA00109521 to 00109525 • 3 ♀♀; same data as for preceding; P. Aguado leg.; DNA CEUA_103; CEUA-CIBIO, CEUA00109526 to 000109527, CEUA00111509 • 4 ♂♂, 2 ♀♀; León, Posada de Valdeón, cerca hotel Cumbres; 43°9′3″ N, 4°55′9″ O; 923 m; 23 August 2021; A. Ricarte and Z. Nedeljković leg.; DNA CEUA_S104; CEUA-CIBIO, CEUA00109528, CEUA00109532, CEUA00111504 • 2 ♂♂, 1♀; León, Posada de Valdeón, Sta. Marina-Prada; 43°8′4.3″ N, 4°53′28.7″ O; 1124 m; 24 August 2021; A. Ricarte and Z. Nedeljković leg.; CEUA-CIBIO, CEUA00109533 to 00109534, CEUA00111505 • 1 ♂; same locality as for preceding; 43°7′57.5″ N, 4°53′19.9″ O; 1138 m; 24 August 2021; A. Ricarte and Z. Nedeljković leg.; CEUA-CIBIO, CEUA00109535 • 1 ♀; León, Posada de Valdeón; 43°9′6.9″ N, 4°54′58.8″ O; 931 m; 27 August 2021; A. Ricarte and Z. Nedeljković leg.; campo de *Daucus*; CEUA-CIBIO, CEUA00109536 • 1 ♂; Madrid, Cercedilla, borde Ctra. M-622; 40°44′10″ N, 4°2′15.7″ O; 1200 m; 27 July 2021; I. Ballester leg.; en *Magydaris panacifolia*; DNA CEUA_S105; CEUA-CIBIO, CEUA00109537 • 1 ♂; same data as for preceding; A. Ricarte leg.; DNA CEUA_S106; CEUA-CIBIO, CEUA00109538 • 1 ♂; same data as for preceding; 28 July 2021; I. Ballester leg.; CEUA-CIBIO, CEUA00109539. ***Revised***. SPAIN • 1 ♂; Ávila, Becedas; 1200 m; 24 August 1980; M. A. Marcos García leg.; CEUA-CIBIO; CEUA00017080; published in Marcos-García [39] • 2 ♂♂; Salamanca, Linares de Riofrío; 10 August 1980; M. A. Marcos García leg.; CEUA-CIBIO, CEUA00017081 to 00017082; published in Marcos-García [39] • 1 ♀; Huesca, Siresa; 20 July 1983; M. A. Marcos García leg.; CEUA-CIBIO, CEUA00017; published in Marcos-García [48] • 1 ♂; Cáceres, Hoyos; 24 May 1980; Marcos García leg.; CEUA-CIBIO, CEUA00017083; published in Marcos-García [39] • 1 ♀; Asturias, Tielve; 10 September 1988; M. A. Marcos García leg.; CEUA-CIBIO, CEUA00017088; published in Marcos-García (1990) • 1 ♀; same locality as for preceding; 21 June 1987; Marcos García leg.; CEUA-CIBIO, CEUA00017084; published in Marcos-García [38] • 1 ♀; Asturias, Santillán; 6 July 1986; M. A. Marcos García leg; CEUA-CIBIO, CEUA00017087; published in Marcos-García [38]. 1 ♂; Girona, Setcases, camino deposito de agua; 1317 m; 30 July 2020; A. Ricarte leg.; CEUA-CIBIO, CEUA00108236; published in Ricarte et al. [53] • 1 ♂; same locality and leg as for preceding; 31 July 2020; CEUA-CIBIO, CEUA00108233; published in Ricarte et al. [53] • 1 ♂, 1 ♀; Girona, Llanars, La Creueta; 1138 m; 29 July 2020; A. Ricarte leg.; CEUA-CIBIO, CEUA00108234, CEUA00108269; published in Ricarte et al. [53] • 1 ♀; same data as for preceding; Z. Nedeljković leg.; CEUA-CIBIO, CEUA00108268; published in Ricarte et al. [53] • 1 ♂; Girona, Villalonga de Ter, Ctra. Villalonga-Setcases, vegetacion cuneta; 1 August 2020; A. Ricarte and Z. Nedeljković leg.; CEUA-CIBIO, CEUA00108235; published in Ricarte et al. [53] • 3 ♀♀; Girona, Setcases, camino del deposito de agua; 1317 m; 31 July 2020; Z. Nedeljković leg.; CEUA-CIBIO, CEUA00108241 to 00108243 • 3 ♀♀; same locality as for preceding; 30 July 2020; Z. Nedeljković leg.; DNA CEUA_S76; CEUA-CIBIO, CEUA00108230 to CEUA00108232 • 1 ♀; same data as for preceding; A. Ricarte leg.; CEUA-CIBIO, CEUA00108240 • 1 ♂, 1 ♀; same locality as for preceding; 31 July 2020; A. Ricarte leg; DNA CEUA_S75; CEUA-CIBIO, CEUA00108237, CEUA00108244; published in Ricarte et al. [53] • 1 ♀; Girona, Villalonga de Ter, camino a la Abella; 1055 m; 3 August 2020; A. Ricarte and Z. Nedeljković leg.; CEUA-CIBIO, CEUA00108267; published in Ricarte et al. [53].

*Additional published records*. PORTUGAL • van Eck [12,13] • SPAIN • Arbucias, Cerdaña, Baños de Montemayor [16]; Leclercq [54]; Kehlmaier [55]; Carles-Tolrá [51].


***Chrysogaster virescens* Loew, 1854**


*Examined material*. ***Revised***. SPAIN • 1 ♂; Ciudad Real, P.N. de Cabañeros; 26 March–14 April 2004; A. Ricarte leg.; malaise trap maJ2; CEUA-CIBIO, CEUA00082503; published in Ricarte [34].

*Additional published records*. SPAIN • Carles-Tolrá and Verdugo [56]; van Eck [57]. All earlier references stated in Ricarte and Marcos-García [14] of *C. virescens* as *Chrysogaster viduata* (Linnaeus, 1758) must be re-located in the checklist, as the name *C. viduata* has often been used in literature for *Melanogaster nuda* [58] but not for *C. virescens*.

#### 3.5.2. *Lejogaster* Rondani, 1857

Below a key to the Iberian species of *Lejogaster*.
Anterior anepisternum entirely bare; male basoflagellomere wider than long … ***L. metallina***
-Anterior anepisternum pilose postero-dorsally; male basoflagellomere as long and wide or slightly longer tan wide … ***L. tarsata***


***Lejogaster metallina* (Fabricius, 1781)**


*Examined material*. ***New***. SPAIN • 2 ♂♂; Huesca, Portalet; 21 June 1988; M. A. Marcos García leg.; CEUA-CIBIO, CEUA00109549 to 00109550 • 1 ♀; Salamanca, cementerio; 9 July 1978; J. J. Pedrero Fdez leg.; CEUA-CIBIO, CEUA00017267 • 1 ♀; Zarzalejos; col. Andréu; UMCZ. ***Revised***. SPAIN • 1 ♀, Cáceres, Cabezabellosa; 6 July 1980; M. A. Marcos García leg.; CEUA-CIBIO, CEUA00017263; published in Marcos-García [39] • 1 ♀; Salamanca, Escurial de la Sierra; 900 m; 10 August 1980; M. A. Marcos García leg.; CEUA-CIBIO, CEUA00017260 • 1 ♀, same locality as for preceding; 15 May 1980; M. A. Marcos García leg.; CEUA-CIBIO, CEUA00017265 • 1 ♀; same locality as for preceding; 18 September 1980; M. A. Marcos García leg.; CEUA-CIBIO, CEUA00017149; published in Marcos-García [39] • 1 ♀; Cáceres, Jerte; 1200 m; 25 August 1980; M. A. Marcos García leg.; CEUA-CIBIO, CEUA00017262, published in Marcos-García [39] • 2 ♂♂; Salamanca, El Maillo; 950 m; 18 June 1980; M. A. Marcos García leg.; CEUA-CIBIO, CEUA00017145 to 00017146; published in Marcos-García [39] • 1 ♂, 1 ♀; Cáceres, El Torno; 950 m; 25 August 1980; M. A. Marcos García leg.; CEUA-CIBIO, CEUA00017143 to 00017144; published in Marcos-García [39] • 1 ♂; Huesca, Aguas Tuertas; 1600 m; 18 July 1983; M. A. Marcos García leg.; CEUA-CIBIO, CEUA00017147; published in Marcos-García [48] • 1 ♂; León, Pto de las Señales; 3 July 1987; M. A. Marcos García leg.; CEUA-CIBIO, CEUA00017148 • 1 ♀; same locality as for preceding; 16 June 1986; M. A. Marcos García leg.; CEUA-CIBIO, CEUA00017266 • 1 ♀; León, Alto de Aralla, 2 June 1988; M. A. Marcos García leg.; CEUA-CIBIO, CEUA00017261; published in Marcos-García [38] • 1 ♀; same locality and leg as for preceding; 14 June 1986; CEUA-CIBIO, CEUA00017264; published in Marcos-García [38] • 1 ♂; Sierra de Guadarrama; 21 June 1927; Dusmet leg.; MNCN, MNCN_Ent 301506; as *Chrysogaster (Liogaster) metallina* in Gil-Collado ([16] • 2 specimens; Villa Rutis; Bolívar leg.; MNCN, MNCN_Ent 301489 to 301490; published in Gil-Collado [16] • 1 specimen; El Burgo; Bolívar; MNCN, MNCN_Ent 301487; published in Gil-Collado [16] • 1 specimen; Coruña; Bolívar leg.; MNCN, MNCN_Ent 301481 • 1 specimen; San Rafael; Gil leg.; MNCN, MNCN_Ent 301480; published in Gil-Collado [16] • 1 specimen; El Burgo; Bolívar leg.; MNCN, MNCN_Ent 301487; published in Gil-Collado [16] • 1 specimen; Parga; Bolívar leg.; MNCN, MNCN_Ent 301488; published in Gil-Collado [16] • 1 specimen; Madrid; Arias leg.; MNCN, MNCN_Ent 301491; published in Gil-Collado [16] • 1 ♂; Sierra de Guadarrama; 25 July 1933; Dusmet leg.; UMCZ; as *Liogaster metallina* in Andréu [52] • 1 ♂; same locality as for preceding; 9 August 1913; Dusmet leg.; UMCZ; as *Liogaster metallina* in Andréu [52] • 1 ♀; same locality as for preceding; 10 April 1931; Dusmet leg.; UMCZ; as *Liogaster metallina* in Andréu [52] • 2 ♀♀; same locality as for preceding; 25 July 1933; Dusmet leg.; UMCZ; as *Liogaster metallina* in Andréu [52] • 1 ♀; same locality as for preceding; 26 June 1935; Dusmet leg.; UMCZ; as *Liogaster metallina* in Andréu [52] • 3 ♀♀; El Pardo; 24 June 1933; Dusmet leg.; UMCZ; as *Liogaster metallina* in Andréu [52].

*Additional published records*. PORTUGAL • van Eck [12] • SPAIN • De la Fuente [59] (as *C. metallina*); Gil-Collado [16] in part (records from Astorga, Cerdaña, Bilbao, Lugo, Oya); Kanervo [60] (as *Liogaster metallina*); van der Goot and Lucas [61]; Pedersen [62]; Maibach and Goeldlin de Tiefenau [15]; Carles-Tolrá [51,63]; Carles-Tolrá et al. [64].


**
*Lejogaster tarsata*
**
**(Meigen, 1822)**


*Examined material. **New.*** SPAIN • 1 ♂; Jaramiel; 21 July 1905; Dusmet leg.; col. Andréu; UMCZ.

*Additional published records*. ANDORRA • van der Goot [50] (as *Liogaster splendida*) • PORTUGAL • van Eck [12,13] • SPAIN • Gil-Collado [16] (as *C. splendida*).

#### 3.5.3. *Melanogaster* Rondani, 1857

Below a key to the Iberian species of *Melanogaster* plus *M. aerosa*, based on Bartsch et al. [44].
Eyes holoptic (males) … 2
-Eyes dichoptic (females) … 5Scutum with sparse pilosity, especially noticeable on the posterior part; surstylus curved and less than two times as long as width … ***M. nuda***
-Scutum with dense pilosity all over (as in Figure 2B,D); surstylus straight or nearly so, more than two times longer than wide (as in Figure 3A) … 3Scutum mainly with brownish black to black hairs; superior lobe of hypandrium with a long tooth pointing opposite the epandrium [7] (figure 4) … ***M. aerosa***
-Scutum mainly with brown to white hairs; superior lobe of hypandrium with a long tooth pointing towards the epandrium (Figure 3A) … 4Scutum with brown hairs; head and thorax without blue reflections; surstylus with a conspicuously elongated finger-like process apically [9] (figure 20) … ***M. hirtella***
-Scutum with white (sometimes yellowish) hairs (Figure 2B); head and thorax usually with blue reflections (Figure 1A,B); surstylus often widened subapically and with a stocky finger-like process apically (Figure 3A) … ***M. baetica* sp. n.**Scutum with very-short sparse pilosity, especially noticeable on the posterior part (hairs shorter than the diameter of a posterior ocellus) … ***M. nuda***
-Scutum with longer dense pilosity all over (hairs longer than the diameter of a posterior ocellus) … 6Scutum with short (length equalling the diameter of an ocellus) semi-recumbent hairs; metatibia with at least some black hairs … ***M. aerosa***
-Scutum with longer (length about twice the diameter of an ocellus) erect or suberect hairs; metatibia completely pale pilose … 7Head and thorax without blue reflections; wing membrane hyaline over all … ***M. hirtella***
-Head and thorax usually with blue reflections (Figure 1C,D); wing usually with a darker area centrally (Figure 1D) … ***M. baetica* sp. n.**


***Melanogaster aerosa* (Loew, 1843)**


(Removed from the Iberian checklist)

*Notes*. We have not found Iberian material of *M. aerosa* amongst all specimens of *Melanogaster* examined. Röder [65] reported *M. aerosa* (as *C. maquarti*) for the first time from Spain with a record from the island of Ibiza. This record, which could not be verified by the authors of the present work, was put in doubt by Kassebeer [66], who indicated that this species is typical from other European climates. Most material of *C. macquarti* reported by Gil-Collado [16] was here reidentified as *C. coerulea* or *M. hirtella*. The specimen of *M. aerosa* reported from Salamanca (Spain) by Carles-Tolrá [51] was reidentified in the present work as *M. hirtella*. Van Eck [12] concluded that a specimen of *M. aerosa* reported from central Portugal by Gomes [67] as *C. macquarti* was actually *M. hirtella*. The external appearances of *M. aerosa* and *M. hirtella* are similar. For example, the scutum of *M. hirtella* has brown hairs that can be as dark as those in *M. aerosa*, but this latter species has more black hairs intermixed with the brown hairs than *M. hirtella*. Genitalia dissection is often required for an accurate diagnosis in these two species. Given that most Iberian records of *M. aerosa* have been reidentified recently or in this work as other brachyopine species, and *M. aerosa* is more typical of northern parts of the continent, we remove this species from the Iberian checklist.


***Melanogaster baetica* sp. n.**


See above, under Section 3.1.


***Melanogaster hirtella* (Loew, 1843)**


*Examined material*. ***New***. SPAIN • 1 ♀; Salamanca, Peñacaballera; 18 May 1980; M. A. Marcos García leg.; CEUA-CIBIO, CEUA00017279 • 1 ♀; Pto Peñanegra, AV; 12 June 1982; C. Urones leg.; CEUA-CIBIO, CEUA00017281 • 3 ♀♀; Pto Peñanegra; same date and leg as for preceding; CEUA-CIBIO, CEUA00025966; CEUA00025971, CEUA00025973 • 1 ♂; San Rafael; C. Bolívar leg.; MNCN, MNCN_Ent 301422; Gil-Collado det. as *C. macquarti* • 1 ♂; same data as for preceding; MNCN, MNCN_Ent 301441; Gil-Collado det. as *C. longicornis* • 1 ♂; Provincia de Madrid; Lauffer leg.; MNCN, MNCN_301464. ***Revised***. SPAIN • 1 ♂; Alberche; Mercet leg.; MNCN, MNCN_Ent 301421; as *C. macquarti* in Gil-Collado [16] • 1 ♂; Salamanca, Villar de Ciervo (Umbegung); 8–18 April 2001; H.-P. Tschorsnig leg.; aus Gelbschalen; MCT; as *M. aerosa* in Carles-Tolrá [51] • 1 ♂, 1 ♀; Salamanca, Béjar, Los Llanos; 1300 m; 30 May 1980; M. A. Marcos García leg.; CEUA-CIBIO, CEUA00017269; published in Marcos-García [39] • 10 ♂♂, 2 ♀♀; Ávila, Cereceda; 1100 m; 4 May 1982; M. A. Marcos García leg.; CEUA-CIBIO, CEUA00017271 to 00017275, CEUA00025974, CEUA00026270 to 00026273, CEUA00017277 to 00017278; published in Marcos-García [39] • 2 ♂♂; Ávila, Becedas; 1200 m; 4 May 1982; M. A. Marcos García leg.; CEUA-CIBIO, CEUA00025969, CEUA00017276; published in Marcos-García [39] • 2 ♀♀; Salamanca, Pañaparda, Dehesa de Perosin; 24 May 1980; M. A. Marcos García leg.; CEUA-CIBIO, CEUA00017282, published in Marcos-García [39] • 1 ♀; Cáceres, El Torno; 950 m; 4 May 1982; M. A. Marcos García leg.; CEUA-CIBIO, CEUA00017284; published in Marcos-García [39] • 1 ♀; Salamanca, Dehesa de Candelario; 9 June 1982; C. Urones leg.; CEUA-CIBIO, CEUA00025975 • 1 ♂; León, Pto Leitariegas; 2 June 1987; M. A. Marcos García leg.; CEUA-CIBIO, CEUA00026242; published in Marcos-García [38] • 1 ♀; León, Leitariegos, Laguna de Arbas; 2 June 1987; M. A. Marcos García leg.; CEUA-CIBIO, CEUA00025977; published in Marcos-García [38] • 3 ♀♀; León, Pto Ancares; 4 July 1987; M. A. Marcos García leg.; CEUA-CIBIO, CEUA00026254 to 00026255, CEUA00026261; published in Marcos-García [38] • 1 ♀; León, Murias de Paredes; 1 June 1988; M. A. Marcos García leg.; CEUA-CIBIO, CEUA00026256; published in Marcos-García [38] • 1 ♂, 1 ♀; León, Beberinos; 2 June 1988; M. A. Marcos García leg.; CEUA-CIBIO, CEUA00026241, CEUA00026266; published in Marcos-García [38] • 2 ♂♂, 3 ♀♀; Alto de Aralla; 1536 m; 3 June 1987; M. A. Marcos García leg.; CEUA-CIBIO, CEUA00026240, CEUA00025978, CEUA00026265, CEUA00026268 to 00026269; published in Marcos-García [38] • 1 ♀; León, Candín; 1 June 1987; M. A. Marcos García leg.; CEUA-CIBIO, CEUA00026257; published in Marcos-García [38] • 1 ♀; León, Valle de Casares; 4 June 1987; M. A. Marcos García leg.; CEUA-CIBIO, CEUA00026252; published in Marcos-García [38] • 1 ♂; León, Pto San Glorio; 1609 m; 22 June 1987; M. A. Marcos García leg.; CEUA-CIBIO, CEUA00026245; published in Marcos-García [38] • 1 ♀; León, Pto del Pando; 14 July 1987; M. A. Marcos García leg.; CEUA-CIBIO, CEUA00026250 • 1 ♀; same data as for preceding; 22 June 1987; CEUA-CIBIO, CEUA00026249 • 1 ♀; León, Morgovejo; 23 June 1987; M. A. Marcos García leg.; CEUA-CIBIO, CEUA00017285; published in Marcos-García [38] • 1 ♀; León, Cofiñal; 3 July 1987; M. A. Marcos García leg.; CEUA-CIBIO, CEUA0002625; published in Marcos-García [38] • 1 ♀; Alicante, Alcoy, Font Roja; 2 June 1994; P. M. Isidro leg.; CEUA-CIBIO, CEUA00026276; published in Isidro [68].

*Additional published records*. PORTUGAL • Gomes [67] (as *Chrysogaster macquarti* = *Melanogaster aerosa*), Lucas [69], van Eck [12,13] • SPAIN • Czerny and Strobl [35]; Gil-Collado [16] (*M. hirtella* var. *claripennis*); Maibach and Goeldlin de Tiefenau [15]; Peréz-Bañón [70].

*Notes*. The Spanish specimens of *M. hirtella* showed some differences with the European *M. hirtella*. For example, the surstylus of the male genitalia was actually more similar in shape to that of *Melanogaster curvistylus* Vujić and Stuke, 1998 [9,71]. It might be that *M. hirtella* var. *claripennis* Strobl *in* Czerny and Strobl, 1909 actually represents a valid species and not a synonym of *M. hirtella*, as it is regarded today [5]. However, we could not collect fresh material of *M. hirtella* from Spain to test whether they were genetically different to the European *M. hirtella*. So, we leave the clarification of the status of the Iberian populations of *M. hirtella* for a future work (Ricarte et al. in prep.), including the revision of the two syntypes of *M. hirtella* var. *claripennis* in P. Strobl’s collection (Admont) [36].


***Melanogaster nuda* (Macquart, 1829)**


*Examined material*. ***New***. SPAIN • 1 ♂; Madrid; Cazurro leg.; MNCN, MNCN_Ent 301462 • 1 ♂; Huesca, Selva de Oza; 18 June 1988; M. A. Marcos García leg.; CEUA-CIBIO, CEUA00109551 • 1 ♀; León, Brañillín; 14 June 1986; M. A. Marcos García leg.; CEUA-CIBIO, CEUA00025958 • 1 ♂; same locality as for preceding; 4 June 1987; M. A. Marcos García leg.; CEUA-CIBIO, CEUA00025837 • 12 ♂♂, 7 ♀♀; León, Pto. Pajares; 4 June 1987; M. A. Marcos García leg.; CEUA-CIBIO, CEUA00025803, CEUA00025796 to 00025799, CEUA00025801 to 00025802, CEUA00025871, CEUA00025822 to 00025825, CEUA00025791 to 00025794, CEUA00025845 to 00025846, CEUA00025850 • 1 ♀; Cofiñal; 3 July 1987; M. A. Marcos García leg.; CEUA-CIBIO, CEUA00025795 • 1 ♂; León, Alto Pontón; 3 June 1988; M. A. Marcos García leg.; CEUA-CIBIO, CEUA00025821 • 1 ♂, 3 ♀♀; Huesca, Selva de Oza; 18 June 1988; M. A. Marcos García leg.; CEUA-CIBIO, CEUA00112365, CEUA00112367 to 00112369 • 3 ♀♀; Huesca, Portalet; 21 June 1988; M. A. Marcos García leg.; CEUA-CIBIO, CEUA00112364, CEUA00112370 to 00112371 • 1 ♀; Huesca, Bujaruelo; 24 June 1988; M. A. Marcos García leg.; CEUA-CIBIO, CEUA00112372 • 1 ♀; Teruel, Frías de Albarracín; 7 July 1997; M. A. Marcos García leg.; CEUA-CIBIO, CEUA00112373. ***Revised***. SPAIN • 2 ♂♂, 6 ♀♀; Murias de Paredes; 1 June 1988; M. A. Marcos García leg.; CEUA-CIBIO, CEUA00025952 to 00025953, CEUA00025880 to 00025884; as *C. lucida* in Marcos-García [38] • 2 ♂♂; León, Geras; 2 June 1988; M. A. Marcos García leg.; CEUA-CIBIO, CEUA00025863 to 00025864; as *C. lucida* in Marcos-García [38] • 19 ♂♂, 20 ♀♀; León, Alto de Aralla; 14 June 1986; M. A. Marcos García leg.; CEUA-CIBIO, CEUA00025819, CEUA00025827 to 00025828, CEUA00025830, CEUA00025832 to 00025836, CEUA00025851 to 00025853, CEUA00025855 to 00025856, CEUA00025858, CEUA00025868, CEUA00025870, CEUA00025887 to 00025888, CEUA00025891, CEUA00025893, CEUA00025899, CEUA00025908, CEUA00025936 to 00025942, CEUA00025944 to 00025951; as *C. lucida* in Marcos-García [38] • 2 ♂♂; same locality as for preceding; 3 June 1987; M. A. Marcos García leg.; CEUA-CIBIO, CEUA00025826, CEUA00025875; as *C. lucida* in Marcos-García [38] • 8 ♀♀; León, Valle de Casares; 4 June 1987; M. A. Marcos García leg.; CEUA-CIBIO, CEUA00025838 to 00025839, CEUA00025840, CEUA00025843, CEUA00025860 to 00025862; as *C. lucida* in Marcos-García [38] • 1 ♂; same data as for preceding; CEUA-CIBIO, CEUA00025874; as *C. lucida* in Marcos-García [38] • 2 ♀♀; León, Hayedo de Geras; 3 June 1987; M. A. Marcos García leg.; CEUA-CIBIO, CEUA00025885 to 00025886, CEUA00025863; as *C. lucida* in Marcos-García [38] • 4 ♂♂, 32 ♀♀; Léon, Pedrosa del Rey; 16 June 1986; M. A. Marcos García leg.; CEUA-CIBIO, CEUA00025788 to 00025790, CEUA00025805 to 00025818, CEUA00025876 to 00025879, CEUA00025892, CEUA00025894, CEUA00025896 to 00025898, CEUA00025900, CEUA00025909, CEUA00025912 to 00025914, CEUA00025955 to 00025957; as *C. lucida* in Marcos-García [38] • 3 ♀♀; León, Morgovejo; 23 June 1987; M. A. Marcos García leg.; CEUA-CIBIO, CEUA00025847 to 00025849; as *C. lucida* in Marcos-García [38] • 1 ♂, 5 ♀♀; León, Murias de Paredes; 1 June 1988; M. A. Marcos García leg.; CEUA-CIBIO, CEUA00025880 to 00025884; as *C. lucida* in Marcos-García [38] • 3 ♂♂, 26 ♀♀; León, Valle de Casares; 14 June 1986; M. A. Marcos García leg.; CEUA-CIBIO, CEUA00025841, CEUA00025873, CEUA00025901 to 00025907, CEUA00025915 to 00025922, CEUA00025924 to 00025934, CEUA00025974; as *C. lucida* in Marcos-García [38] • 16 ♀♀; Huesca, Aguas Tuertas; 1600 m; 18 July 1983; M. A. Marcos García leg.; CEUA-CIBIO, CEUA00025775 to 00025784, CEUA00025959 to 00025964 • 3 ♀♀; same locality as for preceding; 18 July 1983; M. A. Marcos García leg.; CEUA-CIBIO, CEUA00025785 to 00025787; as *C. viduata* in Marcos-García [48].

*Additional published records* (all as *C. viduata*). SPAIN • Leclercq [54]; Marcos-García [48] in part (Candanchú, La Canal Roya, Selva de Oza).

#### 3.5.4. *Orthonevra* Macquart, 1829

Below is a key to the Iberian species of *Orthonevra*. Apart from the Iberian species, the European species *O. incisa* and *O. plumbago* are also included.
Sternum I densely pollinose … 2
-Sternum I shiny or slightly pollinose … 3Body, in general, with shorter hairs (Figure 11A,B); eye virtually bare; katepisternum with two separate patches of hairs, one dorsal and another ventral; metafemur entirely white pilose, at most with some scattered black hairs ventrally (Figure 12C) … ***O. incisa***
-Body, in general, with longer hairs (Figure 4); eye with very short but obvious scattered hairs; katepisternum with a third patch of hairs below the dorsal patch in such a way that dorsal and ventral patches might be connected or almost so; metafemur with black, sometimes spiny, black hairs ventrally … ***O. arcana* sp. n.**Basoflagellomere elongate, 1.5× longer than wide or more … 4
-Basoflagellomere round to oval, less than 1.5× longer than wide … 5Basoflagellomere bar-shaped, with subparallel dorsal and ventral margins … ***O. frontalis***
-Basoflagellomere tapering towards the apex, with a pointing end dorso-apically (more conspicuous in female) … ***O. nobilis***Basoflagellomere 1.3× (Figure 16B). Only females: tergum IV with a small wart-like elevation in the middle of posterior margin; tergum V with a medial sulcus starting on the posterior margin as an incision and then extending almost to the anterior margin as a seam (in posterior view, the sulcus divides the tergum giving it the appearance of two valves) (Figure 17B) … ***O. plumbago***
-Basoflagellomere less than 1.3× longer than wide, usually almost as long as wide. Only females: posterior margin of tergum IV flat, without elevations; posterior margin of tergum V simple, arched in posterior view … 6Basoflagellomere oval, 1.1–1.2× longer than wide; scutum hairs erect; sternum II with erect hairs … ***O. brevicornis***
-Basoflagellomere round, as long and wide or slightly wider than long; scutum hairs recumbent; sternum-II hairs semi-recumbent … ***O. tristis***


***Orthonevra arcana* sp. n.**


See Section 3.1.


***Orthonevra brevicornis* (Loew, 1843)**


*Examined material*. ***New***. SPAIN • 1 ♀; Valencia, Utiel; 9 May 1994; C. Pérez-Bañón leg.; CEUA-CIBIO, CEUA00107987 • 1 ♀; Granada, Sierra Nevada, Monachil, Fuente Alta; 2125 m; 23 June 2021; Z. Nedeljković leg.; CEUA-CIBIO, CEUA00111082. ***Revised***. SPAIN • 1 ♂; Ciudad Real, P. N. Canañeros; 30 May–18 June 2004; A. Ricarte leg.; malaise trap maM2; CEUA-CIBIO, CEUA00086734; published in Ricarte [34] • 1 ♀; Ciudad Real, P. N. Canañeros; 14 April–8 May 2004; A. Ricarte leg.; malaise trap maM1; CEUA-CIBIO, CEUA00086736; published in Ricarte [34] • 1 ♀; Cáceres, La Garganta; 1075 m; 30 May 1980; M. A. Marcos García leg.; CEUA-CIBIO, CEUA00017310; published in Marcos-García [39].

Additional published records. SPAIN • van Eck [57].


**Orthonevra frontalis (Loew, 1843)**


*Examined material*. ***New***. SPAIN • 3 ♂♂, 2 ♀♀; Jaén, P.N. Cazorla, Vadillo Castril, Barranco de la Cabrilla, turbera; 37°55′48.1″ N, 2°46′57.2″ O; 1639 m; 15 June 2019; M. A. Marcos García leg.; DNA CEUA_S63; CEUA-CIBIO, CEUA00107971, CEUA00107989, CEUA00111538, CEUA00111535, CEUA00111540 • 1 ♂, 2 ♀♀; same data as for preceding; A. Ricarte leg.; DNA CEUA_S62; CEUA-CIBIO, CEUA00107970, CEUA00107988, CEUA00111536 • 1 ♂, 1 ♀; Sierra Nevada, Güejar Sierra, El Dornajo, Parking camino a Peña del Perro; 1895 m; 26 June 2021; A. Ricarte leg.; CEUA-CIBIO, CEUA00111541, CEUA00111083 • 1 ♂; same data as for preceding; P. Aguado Aranda leg.; CEUA-CIBIO, CEUA00111542 • 1 specimen; El Escorial; 1 June 1932; Dusmet leg.; MNCN, MNCN_Ent 301523; *Chrysogaster* sp. det. • 1 ♀; El Escorial; 8 June 1935; J. Dusmet leg.; UMCZ. ***Revised***. SPAIN • 2 ♂♂; León, Alto de Aralla; 14 June 1986; M. A. Marcos García leg.; CEUA-CIBIO, CEUA00017312 to 00017313 • 2 ♂♂; same locality as for preceding; 3 June 1987; M. A. Marcos García leg.; CEUA-CIBIO, CEUA00017311, CEUA00017314; published in Marcos-García [38] • 1 ♂, 1 ♀; Ciudad Real, P.N. Cabañeros; 28 May 2004; A. Ricarte leg.; sampling point Pa1; CEUA-CIBIO, CEUA00086737 to 00086738; published in Ricarte [34] • 1 ♂; Ciudad Real, P. N. de Cabañeros; 24 April 2004; S. Rojo leg.; sampling point xPa2; CEUA-CIBIO, CEUA00086739; published in Ricarte [34].

*Additional published records*. ANDORRA • van der Goot [50] (as *Ortoneura frontalis*) • SPAIN • Gil-Collado [16] (as *Chrysogaster frontalis*), Pedersen [62] (as *Orthoneura frontalis*), Marcos-García [72] (as *Orthoneura frontalis*), Anadón [73].


***Orthonevra nobilis* (Fallén, 1817)**


*Examined material*. ***New***. SPAIN • 1 ♂; León, Posada de Valdeón, Santa Marina; 43°7′57.5″ N, 4°53′19.9″ O; 1138 m; 24 August 2021; A. Ricarte and Z. Nedeljković leg.; CEUA-CIBIO, CEUA00111515 • 2 ♂♂, 1 ♀; León, Posada de Valdeón, cerca hotel Cumbres; 43°9′3″ N, 4°55′9″ O; 923 m; 23 August 2021; A. Ricarte and Z. Nedeljković leg.; CEUA-CIBIO, CEUA00111516, CEUA00111523, CEUA00111525 • 1 ♀; León, Posada de Valdeón, Cordiñanes; 43°10′6.5″ N, 4°54′14.8″ O; 843 m; 26 August 2021; Z. Nedeljković leg.; campo de *Daucus*; CEUA-CIBIO, CEUA00111517 • 1 ♀; León, Posada de Valdeón, Cordiñanes, ermita de Corona; 43°10′60″ N, 4°54′9.4″ O; 630 m; 26 August 2021; A. Ricarte leg.; CEUA-CIBIO, CEUA00111518 • 2 ♂♂; León, Posada de Valdeón, Prada, bordes ctra.; 43°8′44.7″ N, 4°54′35.4″ O; 986 m; 25 August 2021; A. Ricarte and Z. Nedeljković leg.; CEUA-CIBIO, CEUA00111519, CEUA00111524 • 1 ♂; León, Posada de Valdeón, Santa Marina; 43°7′57.7″ N, 4°53′19.9″ O; 1138 m; 24 August 2021; A. Ricarte and Z. Nedeljković leg.; CEUA-CIBIO, CEUA00111520 • 1 ♀; León, Posada de Valdeón; 43°9′6.9″ N, 4°54′58.8″ O; 931 m; 27 August 2021; A. Ricarte and Z. Nedeljković leg.; campo de *Daucus*; CEUA-CIBIO, CEUA00111521 • 1 ♀; León, Posada de Valdeón, Prada, bordes cta.; 43°8′44.7″ N, 4°54′35.4″ O; 986 m; 2 August 2021; A. Ricarte and Z. Nedeljković leg.; CEUA-CIBIO, CEUA00111522 • 1 ♂; Jaén, P.N. Cazorla, Vadillo Castril, Barranco de la Cabrilla, turbera; 37°55′48.1″ N, 2°46′57.2″ O; 1639 m; 15 June 2019; A. Ricarte leg.; DNA_CEUA_S57; CEUA-CIBIO, CEUA00107967 • 2 ♂♂; Jaén, P.N. Cazorla, Vadillo Castril, Barranco de la Cabrilla, abrevadero; 37°56′5.18″ N, 2°47′4.9″ O; 1590 m; 15 June 2019; A. Ricarte leg.; CEUA-CIBIO, CEUA00107965 to 00107966 • 2 ♀♀; Teruel, Frías de Albarracín; 7 July 1997; M. A. Marcos García leg.; CEUA-CIBIO, CEUA00107964, CEUA00111543 • 1 ♂; Granada, Sierra Nevada, Monachil, Pradollano, Estación de esquí; 2180 m; 20 June 2021; Z. Nedeljković leg.; CEUA-CIBIO, CEUA00111544 • 1 ♀; Granada, Sierra Nevada, Güejar Sierra, El Dornajo, Parking, camino a Peña del Perro; 1895 m; 24 June 2021; A. Ricarte leg.; CEUA-CIBIO, CEUA00111084 • 2 ♂♂, 4 ♀♀; Madrid, Cercedilla, borde Ctra. M-622; 40°44′10″ N, 4°2′15.7″ O; 1200 m; 27 June 2021; A. Ricarte leg.; en flores de *Magydaris panacifolia*; CEUA-CIBIO, CEUA00111545 to 00111546, CEUA00111548 to 00111551 • 1 ♀; same data as for preceding; I. Ballester leg.; CEUA-CIBIO, CEUA00111547 • 1 ♂; without data; col. Andréu; UMCZ. ***Revised***. SPAIN • 1 ♀; Huesca, Valle de Oza; 1400 m; 18 July 1983; M. A. Marcos García leg.; CEUA-CIBIO, CEUA00017330; published in Marcos-García [48] • 1 ♀; Salamanca, La Alberca; 12 August 1980; M. A. Marcos García leg.; CEUA-CIBIO, CEUA00017327; published in Marcos-García [39] • 1 ♂; Ávila, Becedas; 1200 m; 24 August 1980; M. A. Marcos García leg.; CEUA-CIBIO, CEUA00017318; published in Marcos-García [39] • 1 ♀; Ávila, Cereceda; 1100 m; 4 May 1982; M. A. Marcos García leg.; CEUA-CIBIO, CEUA00017322; published in Marcos-García [39] • 1 ♂; Salamanca, La Honfría; 1100 m; 2 June 1982; M. A. Marcos García leg.; CEUA-CIBIO, CEUA00017321; published in Marcos-García [39] • 1 ♀; Salamanca, Linares de Riofrío; 10 August 1980; M. A. Marcos García leg.; CEUA-CIBIO, CEUA00017320; published in Marcos-García [39] • 1 ♂; same locality and leg as for preceding; 13 June 1979; CEUA-CIBIO, CEUA00017329; published in Marcos-García [39] • 1 ♂; Salamanca, Rinconada de la Sierra; 10 August 1980; M. A. Marcos García leg.; CEUA-CIBIO, CEUA00017319; published in Marcos-García [39] • 1 ♀; Ávila, Tremedal; 13 July 1980; M. A. Marcos García leg.; CEUA-CIBIO, CEUA00017328; published in Marcos-García [39] • 1 ♀; Salamanca, Linares de Riofrio; 10 August 1980; M. A. Marcos García leg.; CEUA-CIBIO, CEUA00017320; published in Marcos-García [39] • 1 ♂; same locality and leg as for preceding; 13 June 1979; CEUA-CIBIO, CEUA00017329; published in Marcos-García [39] • 2 ♂♂, 1 ♀; León, Alto de Aralla; 14 June 1986; M. A. Marcos García leg.; CEUA-CIBIO, CEUA00017324 to 00017326; published in Marcos-García [38] • 1 ♂; León, Valle de Casares; 14 June 1986; M. A. Marcos García leg.; CEUA-CIBIO, CEUA00017323; published in Marcos-García [38] • 1 ♀; León, Morgovejo; 23 June 1987; M. A. Marcos García leg.; CEUA-CIBIO, CEUA00017331; published in Marcos-García [38] • 1 specimen; Madrid (no further data, but this is likely to be the specimen cited in Gil-Collado [16] from Cazurro’s collection); MNCN • 3 ♂♂; Girona, Setcases, camino depósito de agua; 1317 m; 31 July 2020; A. Ricarte leg.; DNA CEUA_S77; CEUA-CIBIO, CEUA00108263 to 00108265; published in Ricarte et al. [53].

*Additional published records.* PORTUGAL • Van Eck [13] • SPAIN • van der Goot and Lucas [61], Leclercq [54], 1♂ from Valle de Oza [48], Kehlmaier [55], Carles-Tolrá [51].


***Orthonevra tristis* (Loew, 1781)**


*Examined material*. ***New***. SPAIN • 2 ♂♂, 2 ♀♀; Huesca, Portalet; 21 June 1988; M. A. Marcos García leg.; CEUA-CIBIO, CEUA00111554 to 00111555, CEUA00107950, CEUA00107958 • 1 ♀; Huesca, v. Tena, Ibon Espelunchiella; 21 June 1988; M. A. Marcos García leg.; CEUA-CIBIO, CEUA00107957 • 1 ♀; Huesca, Canfranc, Candanchú; Dusmet leg.; UMCZ • 1 ♂; León, Puerto Magdalena; 1 June 1988; M. A. Marcos García leg.; CEUA-CIBIO, CEUA00112374. ***Revised*.** SPAIN • 2 ♂♂, 1 ♀; Léon, Alto de Aralla; 1536 m; 3 June 1987; M. A. Marcos García leg.; CEUA-CIBIO, CEUA00107930 to 00107931, CEUA00107945; as *Orthonevra onytes* in Marcos-García [38] • 1 ♂; same locality and leg as for preceding; 2 June 1988; CEUA-CIBIO, CEUA00107948; as *Orthonevra onytes* in Marcos-García [38] • 2 ♀♀; León, Aralla, 3 June 1987; M. A. Marcos García leg.; CEUA-CIBIO, CEUA00107942, CEUA00107951; as *Orthonevra onytes* in Marcos-García [38] • 1 ♂, 1 ♀; León, Brañillin; 2 June 1988; M. A. Marcos García leg.; CEUA-CIBIO, CEUA00107954, CEUA00107946; as *Orthonevra onytes* in Marcos-García [38] • 1 ♂; León, Cofiñal; 3 June 1988; M. A. Marcos García leg.; CEUA-CIBIO, CEUA00107932; as *Orthonevra onytes* in Marcos-García [38] • 1 ♀; León, Alto de Pontón; 6 July 1986; M. A. Marcos García leg.; CEUA-CIBIO, CEUA00107959; as *Orthonevra onytes* in Marcos-García [38] • 4 ♂♂, 2 ♀♀; León, Pto. San Glorio; 6 June 1988; M. A. Marcos García leg.; CEUA-CIBIO, CEUA00107933, CEUA00107939, CEUA00107944, CEUA00107947, CEUA00107955, CEUA00107968; as *Orthonevra onytes* in Marcos-García [38] • 1 ♂; same locality and leg as for preceding; 5 June 1988; CEUA-CIBIO, CEUA00107936; as *Orthonevra onytes* in Marcos-García [38] • 4 ♀♀; same locality and leg as for preceding; 16 July 1987; CEUA-CIBIO, CEUA00107952 to 00107953, CEUA00107940, CEUA00107969; as *Orthonevra onytes* in Marcos-García [38] • 3 ♂♂; León, Hayedo de Pandetrave; 6 June 1988; M. A. Marcos García leg.; CEUA-CIBIO, CEUA00107934, CEUA00107937 to 00107938; as *Orthonevra onytes* in Marcos-García [38] • 2 ♂♂; León, Pto, Pandetrave; 16 July 1987; M. A. Marcos García leg.; CEUA-CIBIO, CEUA00111553, CEUA00107920; as *Orthonevra onytes* in Marcos-García [38] • 1 ♂; same locality and leg as for preceding; 22 June 1987; CEUA-CIBIO, CEUA00107935; as *Orthonevra onytes* in Marcos-García [38] • 1 ♂; same locality and leg as for preceding; 22 June 1987; CEUA-CIBIO, CEUA00107949; as *Orthonevra onytes* in Marcos-García [38] • 1 ♂, 2 ♀♀; León, Pto Pajarez, Brañillin; 4 June 1987; M. A. Marcos García leg.; CEUA-CIBIO, CEUA00107941, CEUA00107943, CEUA00107956; as *Ortnonevra onytes* in Marcos-García [38] • 9 ♂♂, 2 ♀♀; Huesca, Aguas Tuertas; 1600 m; 18 July 1983; M. A. Marcos García leg.; CEUA-CIBIO, CEUA00017335 to 00017345; as *Orthonevra onytes* in Marcos-García [48].

*Notes.* We have revised a male labelled as *O. tristis* from the Illyrian Alps (MNCN), a locality that is relatively close to the type locality of this species in Austria. This male’s genitalia were dissected and found to be similar to those represented in Vujić [9] from an Austrian specimen but also to the Iberian males of *O. onytes* here examined. Some variability was observed in the shape of the superior lobe of the hypandrium and apico-dorsal lobe of aedeagus, but it was regarded as intraspecific variability, because differences did not appear to be constant and never had as much degree of differentiation as amongst the genitalia of other valid species of *Orthonevra*. Differences in characters other than genitalic were not observed between the Illyrian and the Iberian specimens. We did not examine the types of *O. onytes* and *O. tristis*, but some authors [58] suggest *O. onytes*, described from the French Pyrenees and Switzerland, can be a junior synonym of *O. tristis*, which is also assumed to occur in the Pyrenees [9]. For the sake of coherence with other revisions of brachyopines in Europe, e.g., [9], and until the taxonomic status of *O. onytes* is resolved, we prefer to use here the older name *O. tristis* for the Iberian material and delete the name *O. onytes* from the Iberian checklist.

#### 3.5.5. *Riponnensia* Maibach, Goeldlin de Tiefenau and Speight, 1994a

Below a key to the Iberian species of *Riponnensia*.
-Basoflagellomere elongated, over four times longer than wide; antenna black or blackish brown; surstylus of the male genitalia with a small triangular process near the base [6] (figure 18d) … ***R. longicornis***-Basoflagellomere oval, always much less than four times longer than wide; antenna partly orange; surstylus of the male genitalia without such a process [6] (figure 18a) … ***R. splendens***


***Riponnensia longicornis* (Loew, 1843)**


*Examined material*. ***New.*** SPAIN • 1 ♀; Madrid; 13 July 1906; G. Schramm leg.; MNCN, MNCN_Ent 301439 • 2 ♂♂; Bonillo; 1941; col. Andréu; UMCZ • 1 ♂; Aranjuez; 4 June 1913; Dusmet leg.; col. Andréu; UMCZ • 1 specimen; Segovia, La Granja; September 1933; Gil-Collado leg.; MNCN, MNCN_Ent 301522. ***Revised***. SPAIN • 1 ♀; Madrid, El Pardo; 3 September 1905; Arias leg.; MNCN, MNCN_Ent 301440; as *Chrysogaster longicornis* in Gil-Collado ([16] • 1 ♀; Madrid; Andreu leg.; MNCN, MNCM_Ent 301438; as *Chrysogaster longicornis* in Gil-Collado [16] • 1 ♂; Madrid; Bolívar leg.; MNCN, MNCN_Ent 301437; as *Chrysogaster longicornis* in Gil-Collado [16] • 1 ♀; Cáceres, Cabezabellosa; M. A. Marcos García leg.; CEUA-CIBIO, CEUA00017346; published in Marcos-García [39].

*Additional published records*. GIBRALTAR (UK) • Gil-Collado [16] (as *Chrysogaster longicornis-* Algeciras, Madrid, El Escorial, Gibraltar, San Agustín, Madrid (Dusmet)) • PORTUGAL • Lucas [69] (as *Orthonevra longicornis*). SPAIN. Kanervo [60], van der Goot and Lucas [61], Kassebeer [66].


***Riponnensia splendens* (Meigen, 1822)**


*Examined material*. ***New***. SPAIN • 3 ♂♂; Valencia, Sª Mariola, Bocairent, Font del Mas dels Arbres; 3 June 2020; A. Ricarte leg.; campo *Thapsia villosa*; DNA CEUA_S55; CEUA-CIBIO, CEUA00107924, CEUA00108501 to 00108502 • 1 ♂; same data as for preceding; Z. Nedeljković leg.; CEUA-CIBIO, CEUA00108500 • 1 ♀; same locality as for preceding; 12 June 2021; A. Ricarte leg.; campo *Thapsia villosa*; CEUA-CIBIO, CEUA00111526 • 1 ♂; Alicante, Alcoleja, Sierra Aitana, Puerto Tudons; 6 June 2020; Z. Nedeljković leg.; campo de *Thapsia villosa*; CEUA-CIBIO, CEUA00107923 • 1 ♂; same locality and leg as for preceding; 26 May 2021; DNA CEUA_S58; CEUA-CIBIO, CEUA00107925 • 1 ♂; León, Posada de Valdeón; 43°9′6.9″ N, 4°54′58.8″ O; 931 m; 25 August 2021; A. Ricarte and Z. Nedeljković leg.; campo de *Daucus*; CEUA-CIBIO, CEUA00111527 • 1 ♂; Madrid, Cercedilla, borde Ctra. M-622; 40°2′15.7″ N, 4°2′15.7″ O; 1200 m; 27 July 2021; A. Ricarte leg.; CEUA-CIBIO, CEUA00111531 • 3 ♂♂; same data as for preceding; I. Ballester leg.; CEUA-CIBIO, CEUA00111528 to 00111530 • 1 ♂, 1 ♀; Granada, Sierra Nevada, Padul, Lagunas de Padul; 726 m; 25 June 2021; A. Ricarte leg.; CEUA-CIBIO, CEUA00111532 to 00111533 • 1 ♂; Jaén, La Iruela, Arroyo del Valle meadow, next to A-319 road; 37°55′12.54″ N, 2°56′57.37″ W; 1006 m; 31 May 2018; A. Ricarte leg.; with flowering *Crataegus* sp.; CEUA-CIBIO, CEUA00107921 • 1 ♂; same data as for preceding; 1 June 2018; DNA CEUA_S56, INV09452; CEUA-CIBIO, CEUA00107922 • 1 ♀; Salamanca, Zarapicos; 26 May 1989; M. A. Marcos García leg.; CEUA-CIBIO, CEUA00111534 • 1 ♂; Salamanca, Aldealungua; 27 June 1979; M. A. Marcos García leg.; CEUA-CIBIO, CEUA00017347 • 1 ♂; Salamanca, Cabrerizos; 17 September 1978; M. A. Marcos García leg.; CEUA-CIBIO, CEUA00017352 • 1 ♂; Badajoz, Mérida, Embalse de Cornalvo; 38°59′ N, 6°10′ W; 13 April 2000; W. van Steenis and E. S. Bakker leg.; CEUA-CIBIO, CEUA00086745 • 1 ♂; Huelva, Santa Olalla del Cala, Dehesa de San Francisco (*Q. rotundifolia*), cerca casa recepción visitantes en flores tipo *Thapsia*; 37.87843, -6.23621; 21 May 2016; A. Ricarte leg.; CEUA-CIBIO, CEUA00106343 • 1 ♀; Granada, Alhama; 800 m; June 1942; Dusmet leg.; UMCZ • 1 ♂; Madrid, Galapagar; 18 July 1932; Dusmet leg.; UMCZ. ***Revised***. MOROCCO • 1 ♂; Tanger; M. Escalera leg; MNCN, MNCN_Ent 301436; as *Chrysogaster splendens* in Gil-Collado [16] • SPAIN • 2 ♀♀; Asturias, Tielve; 21 June 1987; M. A. Marcos García leg.; CEUA-CIBIO, CEUA00017359 to 00017360; published in Marcos-García [38] • 1 ♀; Asturias, Sotres; 5 June 1988; M. A. Marcos García leg.; CEUA-CIBIO, CEUA00017361; published in Marcos-García [38] • 1 ♂; Sierra de Guadarrama; 18 September 1911; Dusmet leg.; published in Andréu [52] • 1 ♂; Ciudad Real, P.N. Cabañeros; 24 August–12 Sept. 2004; A. Ricarte leg.; malaise trap maJ1; CEUA-CIBIO, CEUA00086743; published in Ricarte [34] • 1 ♂; Valencia, Requena; 5 September 1993; C. Pérez-Bañon leg.; CEUA-CIBIO, CEUA00017355; published in Pérez-Bañón [70] • 1 specimen; El Pardo; 25 July 1904; Arias leg.; MNCN, MNCN_Ent 301425; as *Chrysogaster splendens* in Gil-Collado [16] • 1 specimen; same locality and leg as for preceding; 3 June 1906; MNCN, MNCN_Ent 301428; as *Chrysogaster splendens* in Gil-Collado [16] • 1 specimen; same locality and leg as for preceding; 3 September 1905; MNCN, MNCN_Ent 301426; as *Chrysogaster splendens* in Gil-Collado [16] • 1 specimen; same locality and leg as for preceding; 17 June 1906; MNCN, MNCN_Ent 301427 as *Chrysogaster splendens* in Gil-Collado [16] • 1 specimen; Villaviciosa de Odón; Ardois leg.; MNCN, MNCN_Ent 301429; as *Chrysogaster splendens* in Gil-Collado [16] • 1 ♂; Salamanca, Santibáñez de la Sierra; 24 September 1980; M. A. Marcos García leg.; CEUA-CIBIO, CEUA00017358; as *Orthonevra frontalis* in Marcos-García [39] • 1 ♀; Ávila, Becedas; 1200 m; 30 September 1980, M. A. Marcos García leg.; CEUA-CIBIO, CEUA00017357; as *Orthonevra frontalis* in Marcos-García [39] • 1 ♂; Salamanca, El Cabaco; 29 June 1980; M. A. Marcos García leg.; CEUA-CIBIO, CEUA00017351; as *Orthoneura frontalis* in Marcos-García [39] • 1 ♂; Salamanca; Escurial de la Sierra; 18 September 1980; M. A. Marcos García leg.; CEUA-CIBIO, CEUA00017362; as *Orthoneura frontalis* in Marcos-García [39] • 1 ♂; Cáceres, Gata; 27 September 1980; M. A. Marcos García leg.; CEUA-CIBIO, CEUA00017354 • 1 ♀; same locality and leg as for preceding; 3 July 1980; CEUA-CIBIO, CEUA00017363; as *Orthoneura frontalis* in Marcos-García [39] • 1 ♂; Cáceres, Jerte; 600 m; 25 August 1980; M. A. Marcos García leg.; CEUA-CIBIO, CEUA00017349; as *Orthoneura frontalis* in Marcos-García [39] • 1 ♂; Salamanca, Linares de Riofrío; 17 June 1980; M. A. Marcos García leg.; CEUA-CIBIO, CEUA00017353; as *Orthoneura frontalis* in Marcos-García [39] • 1 ♀; Salamanca, El Maíllo; 925 m; 13 July 1980; M. A. Marcos García leg.; CEUA-CIBIO, CEUA00017356; as *Orthoneura frontalis* in Marcos-García [39] • 1 ♂; Cáceres, Nuñomoral; 28 September 1980; M. A. Marcos García leg.; CEUA-CIBIO, CEUA00017348; as *Orthoneura frontalis* in Marcos-García [39] • 1 ♂; Salamanca, Serradilla del Llano; 28 September 1980; M. A. Marcos García leg.; CEUA-CIBIO, CEUA00017350; as *Orthoneura frontalis* in Marcos-García [39].

*Additional published records.* PORTUGAL • See checklist in van Eck [12] • SPAIN • Gil-Collado [16] (-partly Algeciras, El Escorial, San Agustín, El Escorial, Sierra de Guadarrama, Borovia, Astorga, Moya, Viladrau, Túy, Centellas); van der Goot and Lucas [61]; Maibach and Goeldlin de Tiefenau [15]; Kassebeer [66]; Carles-Tolrá [51]; van Eck [57].

## 4. Molecular Study

A total of 23 brachyopine species were included in the molecular analyses: 1 of *Brachyopa*, 3 of *Chrysogaster*, 2 of *Lejogaster*, 4 of *Melanogaster* and 14 of *Orthonevra*. We were able to obtain sequences for 12 of the 25 specimens whose DNA was isolated. This is the first time that COI-5’ barcodes for *C. coerulea* and *O. frontalis* are published; we also obtained new sequences for *C. solstitialis* and *O. nobilis*. Both NJ and ML trees had virtually the same topologies. In the ML tree (Figure 18), four main clades were obtained with the following node support values (ns): *Orthonevra* (ns = 76), *Melanogaster* (ns = 94), *Chrysogaster* (ns = 86) and *Lejogaster* (ns = 99). The new species, *M. baetica* sp. n. and *O. arcana* sp. n., conform single clades with bootstrap values of 94 and 97, respectively. The estimation of evolutionary divergence of COI-5′ sequences revealed high values of interspecific variability (>3%) in *Chrysogaster* and *Orthonevra* but low (<3%) in *Lejogaster* and *Melanogaster* (see Appendix A).

## 5. Discussion

After the present work, the total number of species of the five studied genera of Brachyopini grows from 15 to 18 in the Iberian–Balearic region: *Chrysogaster*, 6 spp.; *Lejogaster*, 2 spp.; *Melanogaster*, 3 spp.; *Orthonevra*, 5 spp.; *Riponnensia*, 2 spp. The small size of these hoverflies, their similar overall appearance and look when flying in the field, and the difficulty to identify females—and males, in the absence of dissected genitalia—have limited the number of studies on their regional faunas and taxonomy. Although various recent monographs include keys to genera/species (e.g., van Veen [43], Bartsch et al. [44]), the present work is the second after Vujić [9] devoted at the regional revision of the above-mentioned Brachyopini genera in Europe and the first in revising their species diversity in the south-west of the continent.

The distribution of many brachyopines is now better understood in the Iberian region as a result of the present work. For example, *Chrysogaster basalis*, which was thought to be present in various Spanish provinces from the very south (Cádiz) to the very north (Huesca) [14], is now confirmed only from the northern provinces of Huesca and Teruel, suggesting that the Iberian distribution of the actual *C. basalis* is more limited than previously expected to be. Another interesting example is *Lejogaster tarsata*. The only (and old) Spanish record of this species was from the Girona province [16]. In the mid-twentieth century, it was reported from Andorra [50], and it was just recently incorporated to the Portuguese checklist of hoverflies [12]. The disperse distributional record of this species suggested that it should occur in a wider area, and, in fact, we found an old specimen (year 1905) collected in Jaramiel (Valladolid, Spain), somewhere in between Girona and Portugal. At the European level, the finding of *C. rondanii* is especially important, since this species had been confirmed only from the Netherlands, Germany, Belgium, France, Czechia and Switzerland [74]. The male of *C. rondanii* reported from Ávila represents the southernmost record of this species in Europe.

The results obtained for the genus *Chrysogaster* address the fact that synonyms are sometimes proposed without a detailed taxonomic analysis behind. The Spanish endemic variety *coerulea* of *C. chalybeata* was proposed by Strobl *in* Czerny and Strobl [35]. Authors such as Andréu [52] and Gil-Collado [16] retained the use of this variety name. After a long temporal gap, this variety was upgraded to the species level and given as a synonym of *C. chalybeata* in the catalogue by Peck [5]. Works of reference in brachyopines, e.g., [6], did not mention this variety. However, the finding of males with blue shine and distinctive genitalia (Figure 9) prompted us to search the literature, and this led to the conclusion that these males coincided with Strobl’s concept of *coerulea*. This is not the only example of species incorrectly synonymised in Syrphidae, as indicated by the results of studies such as Nedeljković et al. [75] and Marcos-García et al. [76]. *Chrysogaster coerulea* separates clearly in the COI-based tree from all other species of *Chrysogaster* analysed (Figure 18), confirming that the morphological singularity of Spanish populations of *Chrysogaster* described by Strobl in 1909 addressed to a valid taxon. Additionally, within *Chrysogaster*, *C. solstitialis* appears to represent a single taxon from the morphological and molecular point of view (Figure 18), from the Iberian Peninsula to Scandinavia, all across Europe. 

The *Melanogaster* study resulted in the description of *M. baetica* sp. n. from Spain and the elimination of *M. aerosa* from the Iberian–Balearic checklist of Syrphidae. This latter species is confirmed from central and northern Europe and, further south, only from high mountains of Montenegro [9]. Most specimens of *M. aerosa* (also as *C. macquarti* in literature) from the Iberian region were revised and confirmed to belong to other *Melanogaster* species, so it is just plausible that *M. aerosa* is really absent from this southern region of Europe. However, the related species *M. baetica* sp. n. is well represented in the Baetic mountain system of Spain. This new Spanish endemic species is morphologically most similar to *M. parumplicata*, which is confirmed from central and northern Europe but also from mountains of France, Switzerland, Bosnia and Herzegovina and Montenegro. The new *Melanogaster* species is also similar to *M. hirtella*, but the male of this latter species is unique in having a conspicuously elongated apex of the surstylus, much shorter and stockier in *M. baetica* sp. n. (Figure 3A); *M. hirtella* also has darker (brown) hairs on the scutum than *M. baetica* sp. n., which has white to yellowish white hairs. *Melanogaster baetica* sp. n. may have evolved in isolation in the Iberian Peninsula, where the close *M. parumplicata* may also be present in high northern mountains such as the Pyrenees, but unexpected further south. *Melanogaster aerosa*, *M. baetica* sp. n., *M. parumplicata* and the North African species *M. lindbergi* appear to belong to the same species group due to their similar morphology, especially in their male genitalia [7,8,9]. From the molecular point of view, *M. hirtella*, with slightly more differentiated male genitalia, would also belong to the same clade as the above-mentioned species (Figure 18).

The better understanding of the *O. incisa* concept as well as the description of *O. arcana* sp. n. confirm that a shiny or slightly pollinose sternum I cannot be taken as diagnostic for *Orthonevra*, because these two species have a wholly pollinose sternum I, unlike all other European species of this genus [6]. *Orthonevra arcana* sp. n. and *O. incisa* share the pollinose sternum I with *Chrysogaster*/*Melanogaster*, but *Orthonevra* and *Chrysogaster*/*Melanogaster* differ in various other morphological characters. Curiously, Moran et al. [11] recovered the genera *Orthonevra* and *Chrysogaster* within the same molecular clade. A new group, the *O. incisa* group, is morphologically defined in this preliminary study for species sharing a pollinose sternum I, females with tergum 5 triangular in dorsal view and with a posterior incision (of variable size), and males with asymmetric superior lobes of the aedeagus cover. The specimen of *O. frontalis* is close to *O. tristis* and *O. arcana* sp. n. in the tree, but this grouping might just be accidental, since the bootstrap support for the clade containing all the three species is 51 (Figure 18). Fresh material of *O. incisa* and additional molecular markers are necessary to test the putative relationship between this species and *O. arcana* sp. n.. Thus, a new gene-based tree would be expected to show *O. arcana* sp. n. and *O. incisa* grouping together in the same clade.

Molecular analyses in the present work indicate that the species of some of the studied genera differ clearly in the COI-5′ marker, while those of other genera have a lower degree of differentiation in this marker but are still different morphologically. For example, the divergence values amongst morphologically distinct species of *Orthonevra* range from 3–21%, while amongst species of *Melanogaster*, they are just up to 1% (see Appendix A). Despite the fact that more than 90% of congeneric species in the Diptera have high values of divergence in the COI-5′ barcodes, there is a 3% of cases for which the divergence is low between valid species [77]. A similar morphology combined with low genetic divergence among species are clear indicators of closely related species [78]. Furthermore, the low genetic divergence between congeneric species in *Lejogaster* and *Melanogaster* may suggest that these genera have been originated in a recent speciation process [79]. Nevertheless, the morphological and molecular data supported our species concepts and confirmed the validity of the two new species and *C. coerulea*. This differential variation in the COI-5′ for genera that are now affiliated to Brachyopini may indicate that this tribe is in fact paraphyletic, in accordance with the findings by Moran et al. [11]. Therefore, a complete sampling together with the use of further molecular markers would be necessary to test phylogenetic relationships in those genera involving less-differentiated species such as those of *Melanogaster* or *Lejogaster* and more-differentiated as in *Chrysogaster* and *Orthonevra*.

Brachyopines of the five studied genera are hoverflies with aquatic or semiaquatic saprophagous larvae found in plant-rich mediums. Thus, adults are usually observed near small water courses, ponds, and bogs [80], as confirmed in our study with many localities including aquatic habitats (e.g., ‘La Font del Mas dels Arbres’ and ‘Ibon Espelunchiella’). The habitats of the studied brachyopines are especially vulnerable to droughts and changes of any sort in their water regimes. In fact, some of the studied species are catalogued under a threatened category in Europe, *C. basalis* as Vulnerable (VU) [81], *C. rondanii* as Endangered (EN) [74] and *R. longicornis* also as Endangered (EN) [82]. Special attention should be paid to these latter two species and a better assessment of their population trends and specific threats in the Iberian Peninsula should also be undertaken for their conservation.

## 6. Conclusions

Morphological and molecular evidence was combined to update the taxonomy and systematics of Iberian brachyopines. According to the main aim of this work, the conclusions are as follows:(1)A total of 18 species of *Chrysogaster*, *Lejogaster*, *Melanogaster*, *Orthonevra* and *Riponnensia* (Brachyopini) are present in the Iberian–Balearic Region. *Chrysogaster coerulea* is reinstated as a valid species and *M. aerosa* is removed from the Iberian checklist. A lectotype is designated for the non-Iberian species *Orthonevra plumbago*.(2)Two new Spanish endemic species, *M. baetica* sp. n. and *O. arcana* sp. n., are described and illustrated.(3)From the molecular point of view, high levels of interspecific variability in COI-5′ were revealed for *Chrysogaster* and *Orthonevra*, but low levels for *Lejogaster* and *Melanogaster*.

This is the first work devoted to update the taxonomy, systematics, and distribution of Brachyopini in the Iberian area.

## Figures and Tables

**Figure 1 insects-13-00648-f001:**
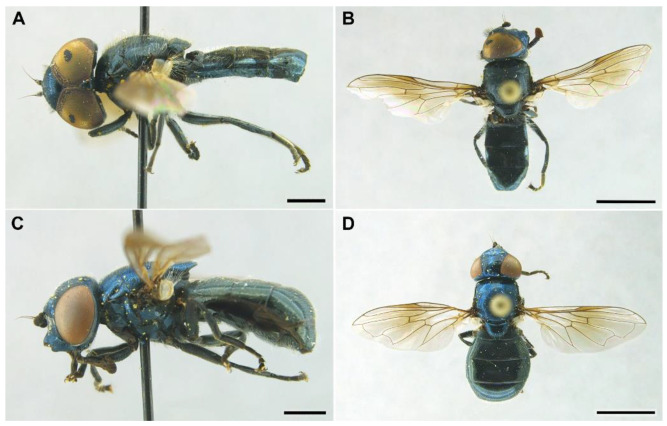
*Melanogaster baetica* sp. n., overall appearance. (**A**,**B**): Male holotype. (**C**,**D**): Female paratype. (**A**,**C**): Lateral view. (**B**,**D**): Dorsal view. Scale bars = (**A**,**C**): 1 mm; (**B**,**D**): 2 mm.

**Figure 2 insects-13-00648-f002:**
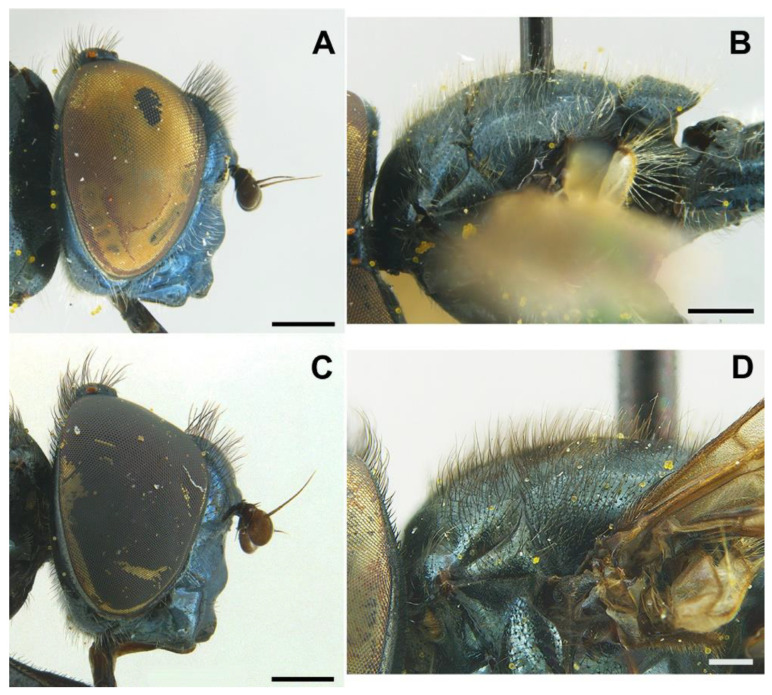
Males of *Melanogaster*. (**A**,**B**): *Melanogaster baetica* sp. n., holotype. (**C**,**D**): *Melanogaster parumplicata* from Norway. (**A**,**C**): Head, lateral view. (**B**,**D**): Thoracic scutum, lateral view. Scale bars = (**A**–**C**): 0.5 mm; (**D**): 0.25 mm.

**Figure 3 insects-13-00648-f003:**
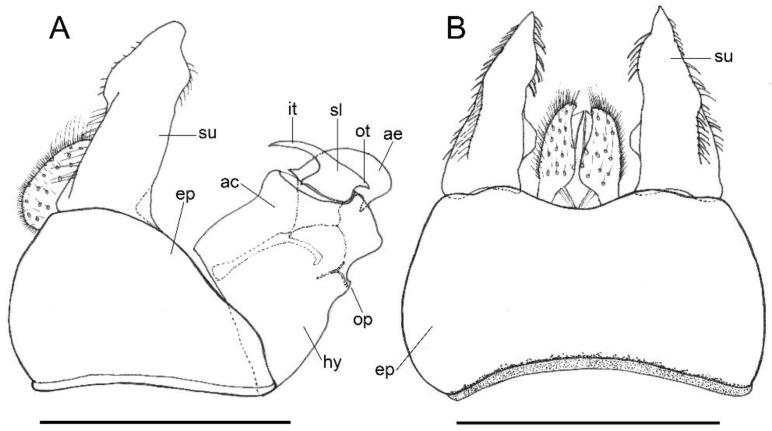
Male of *Melanogaster baetica* sp. n., genitalia (adapted from holotype). (**A**): Lateral view (left side). (**B**): Dorsal view. Legend: ac, aedeagus cover; ae, aedeagus; ep, epandrium; hy, hypandrium; it, inner tooth of the superior lobe of the aedeagus cover; op, outer projection of the hypandrium; ot, outer tooth of the superior lobe of the aedeagus cover; sl, superior lobe of the aedeagus cover; su, surstylus. Scale bars = 0.5 mm.

**Figure 4 insects-13-00648-f004:**
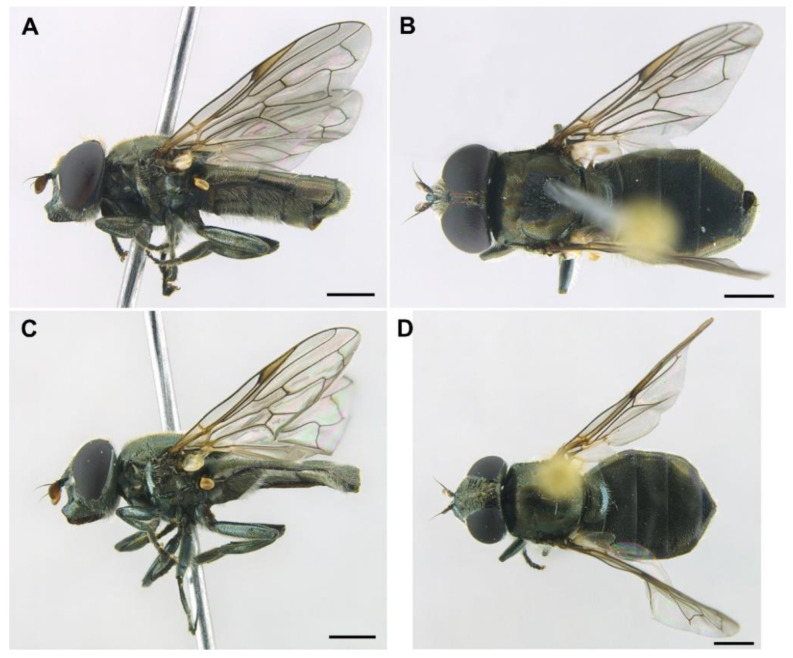
*Orthonevra arcana* sp. n., overall appearance. (**A**,**B**): Male holotype. (**C**,**D**): Female paratype. (**A**,**C**): Lateral view. (**B**,**D**): Dorsal view. Scale bars = 1 mm.

**Figure 5 insects-13-00648-f005:**
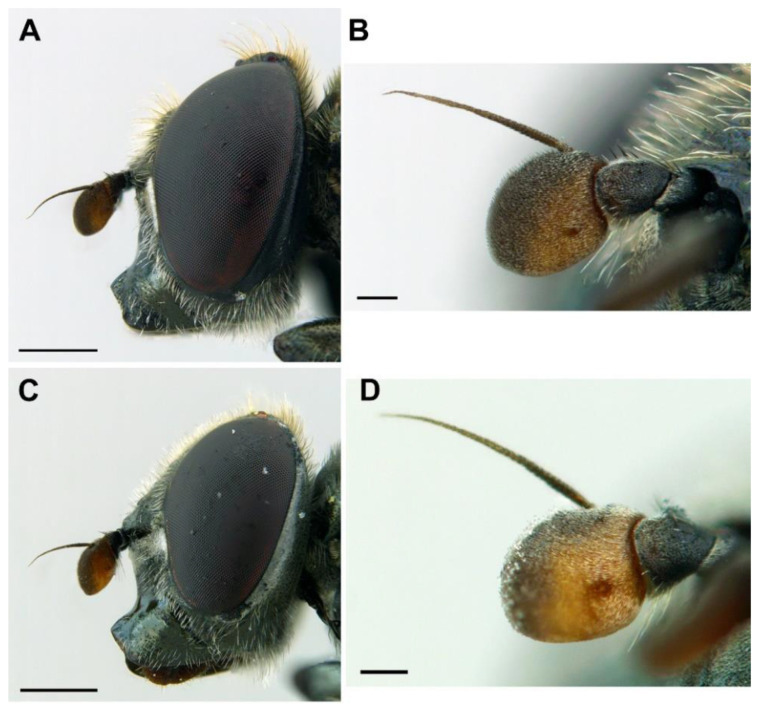
*Orthonevra arcana* sp. n. (**A**,**B**): Male holotype. (**C**,**D**): Female paratype. (**A**,**C**): Head, lateral view. (**B**,**D**): Right antenna, lateral view. Scale bars = (**A**,**C**): 0.5 mm; (**B**,**D**): 0.1 mm.

**Figure 6 insects-13-00648-f006:**
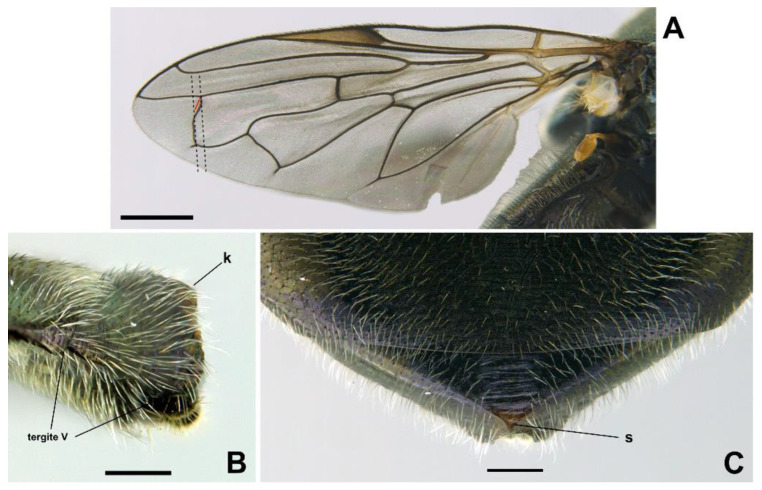
*Orthonevra arcana* sp. n. (**A**): Male holotype. (**B**,**C**): Female paratype. (**A**): Right wing. (**B**)**:** Abdomen apex, lateral view. (**C**): Abdomen apex, dorsal view. Legend: k—keel, s—spina-like process. Red arrow “a” indicates that vein M_1_ is recessive, i.e., the intersection M_1_/R_4+5_ is closer to the wing base. Scale bars = (**A**): 0.75 mm; (**B**,**C**): 0.25 mm.

**Figure 7 insects-13-00648-f007:**
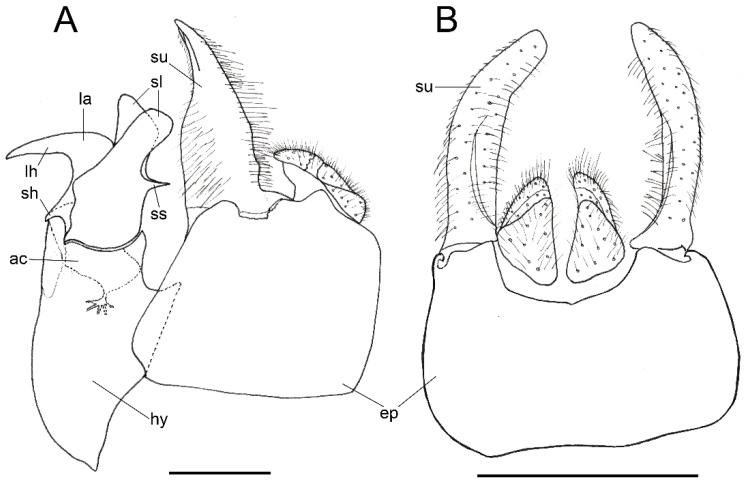
*Orthonevra arcana* sp. n., male genitalia (adapted from holotype). (**A**): Lateral view (right side). (**B**): Dorsal view. Legend: ac, aedeagus cover; ep, epandrium; hy, hypandrium; la, apico-dorsal lobe of aedeagus; lh, large hook-like process of the apico-dorsal lobe of aedeagus; sh, small hook-like process of the apico-dorsal lobe of aedeagus; sl, superior lobe of the aedeagus cover; ss, spur of the superior lobe of the aedeagus cover; su, surstylus. Scale bars = (**A**): 0.25 mm; (**B**): 0.5 mm.

**Figure 8 insects-13-00648-f008:**
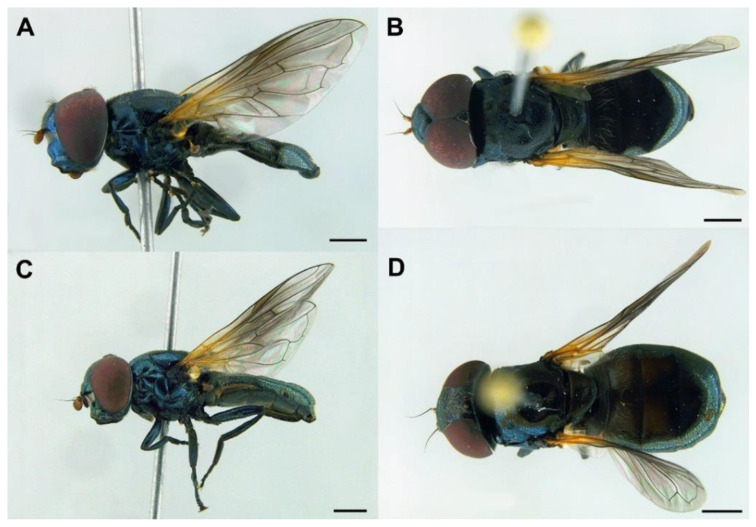
*Chrysogaster coerulea*, overall appearance. (**A**,**B**): Male. (**C**,**D**): Female. Lateral (**A**,**C**): view. (**B**,**D**): Dorsal view. Scale bars = 1 mm.

**Figure 9 insects-13-00648-f009:**
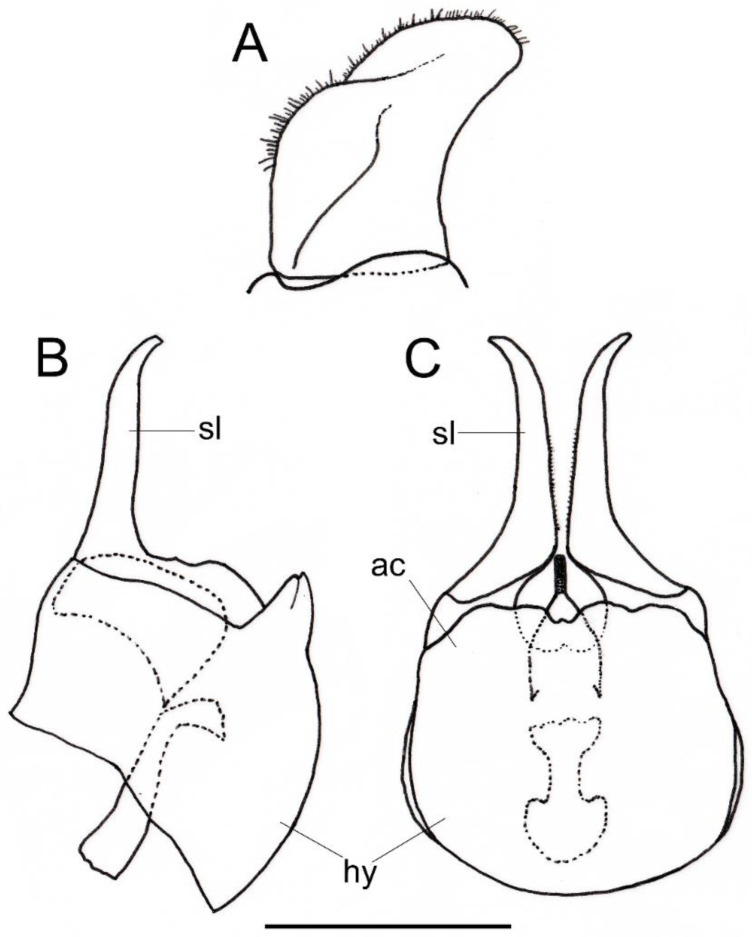
*Chrysogaster coerulea*, male, genitalia. (**A**): Left surstylus, lateral view. (**B**): Hypandrium, lateral view. (**C**): Hypandrium, ventral view. Legend: ac, aedeagus cover; hy, hypandrium; sl, superior lobe of the aedeagus cover. Scale bars = 1 mm.

**Figure 10 insects-13-00648-f010:**
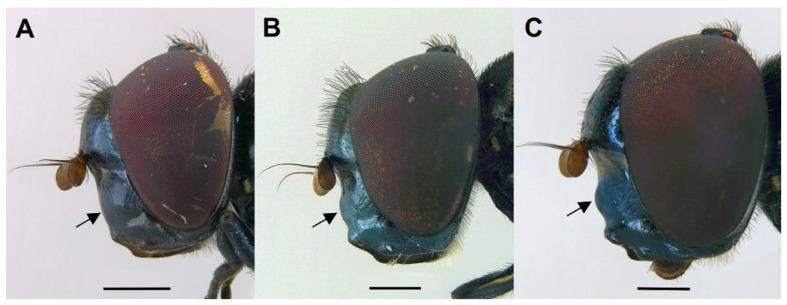
Males of *Chrysogaster*, head, lateral view. (**A**): *Chrysogaster basalis*. (**B**): *Chrysogaster solstitialis*. (**C**): *Chrysogaster coerulea*. An arrow indicates the position of the facial tubercle. Scale bars = (**A**): 0.75 mm; (**B**,**C**): 0.5 mm.

**Figure 11 insects-13-00648-f011:**
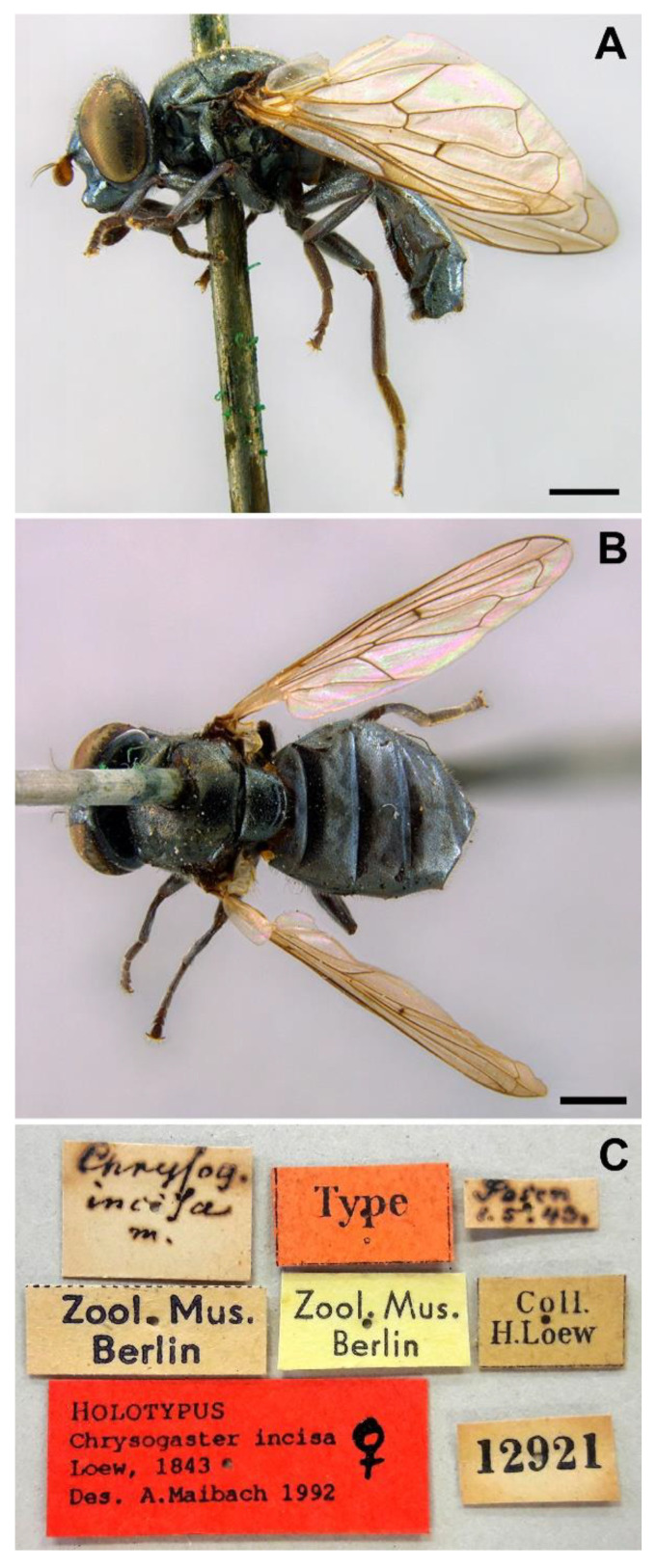
*Orthonevra incisa*, female holotype. (**A**): Overall appearance, lateral view. (**B**): Overall appearance, dorsal view. (**C**): Original labels. Scale bars = 1 mm.

**Figure 12 insects-13-00648-f012:**
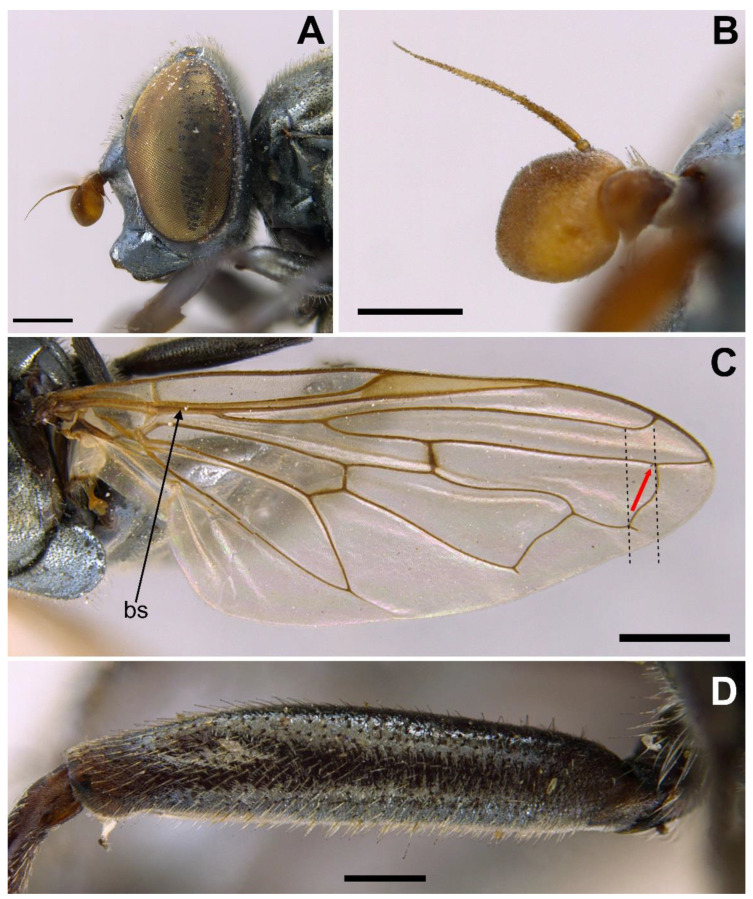
*Orthonevra incisa*, female holotype. (**A**): Head, lateral view. (**B**): Right antenna, inner side, lateral. (**C**): Right wing. (**D**): Left metafemur. Red arrow indicates that vein M_1_ is not recessive, i.e., the intersection M_1_/R_4+5_ is closer to the wing apex. Legend: bs, basal section of radial vein. Scale bars = (**A**): 0.5 mm; (**B**,**D**): 0.25 mm; (**C**): 1 mm.

**Figure 13 insects-13-00648-f013:**
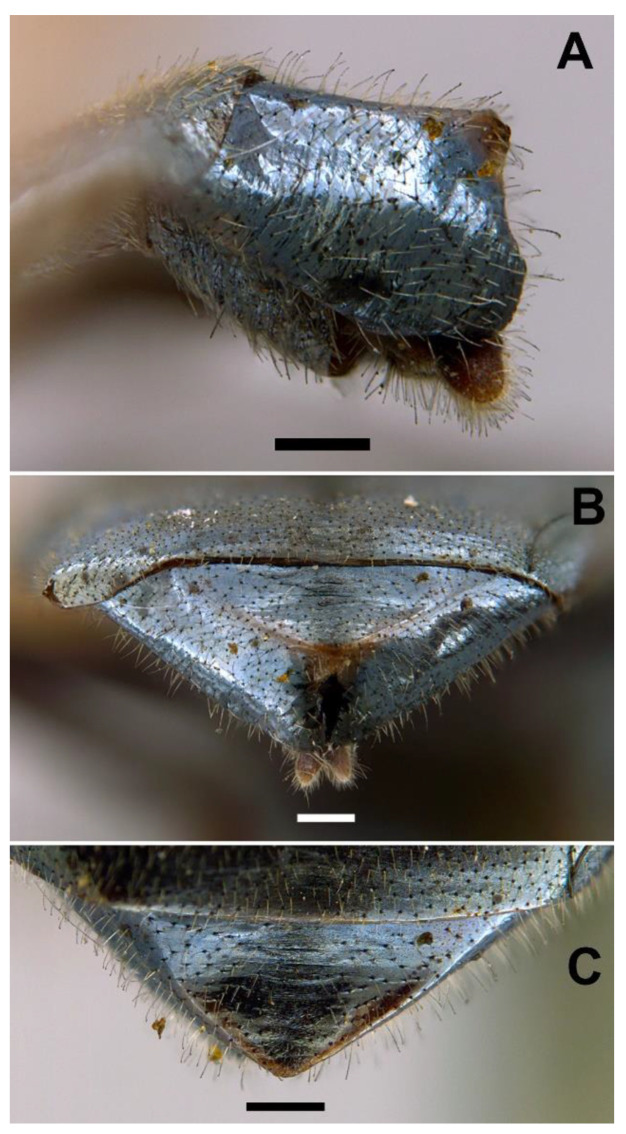
*Orthonevra incisa*, female holotype, abdomen apex. (**A**): Lateral view. (**B**): Posterior view. (**C**): Dorsal view. Scale bars = 0.25 mm.

**Figure 14 insects-13-00648-f014:**
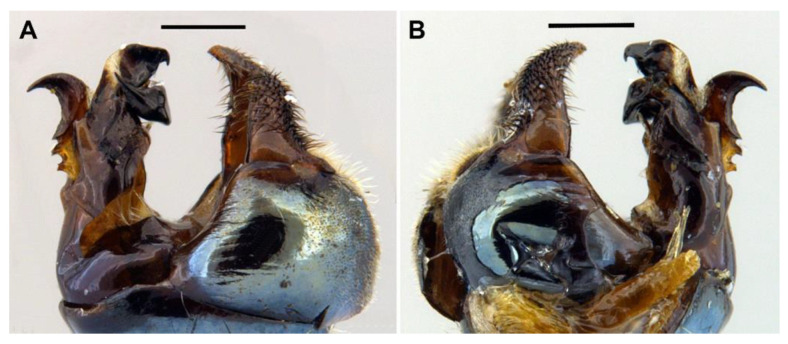
*Orthonevra incisa* from Poland, male, genitalia, lateral view. (**A**): Right side. (**B**): Left side. Scale bars = 0.25 mm.

**Figure 15 insects-13-00648-f015:**
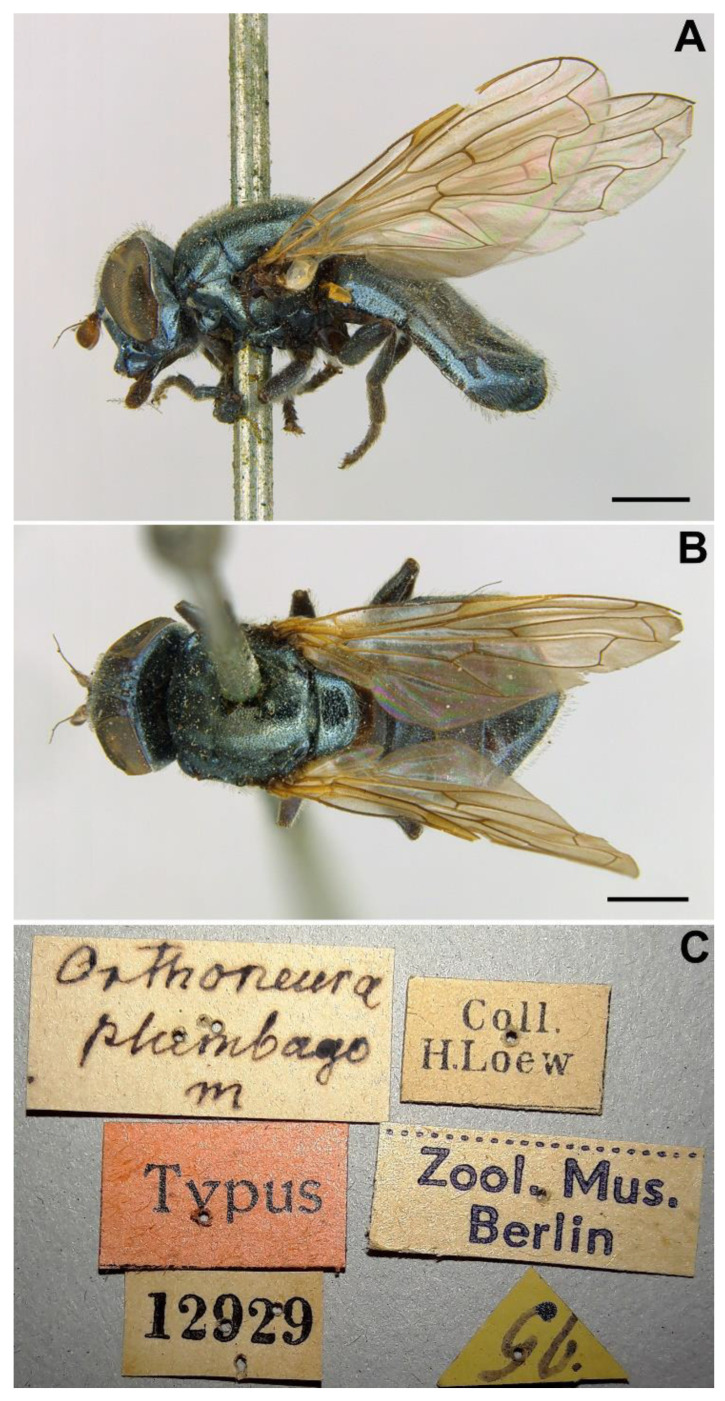
*Orthonevra plumbago*, female lectotype. (**A**): Overall appearance, lateral view. (**B**): Overall appearance, dorsal view. (**C**): Original labels. Scale bars = 1 mm.

**Figure 16 insects-13-00648-f016:**
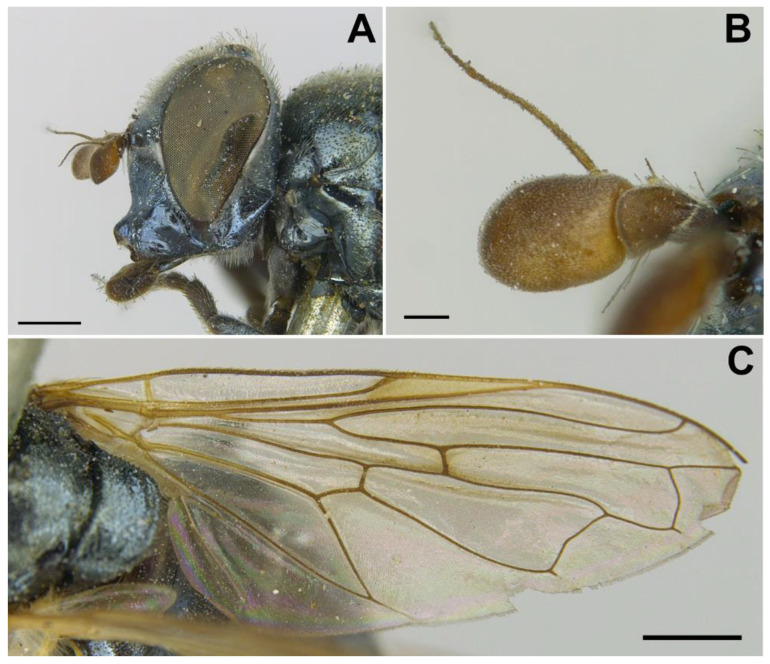
*Orthonevra plumbago*, female lectotype. (**A**): Head, lateral view. (**B**): Right antenna, inner side, lateral. (**C**): Right wing. Scale bars = (**A**): 0.5 mm; (**B**): 0.1 mm; (**C**): 0.75 mm.

**Figure 17 insects-13-00648-f017:**
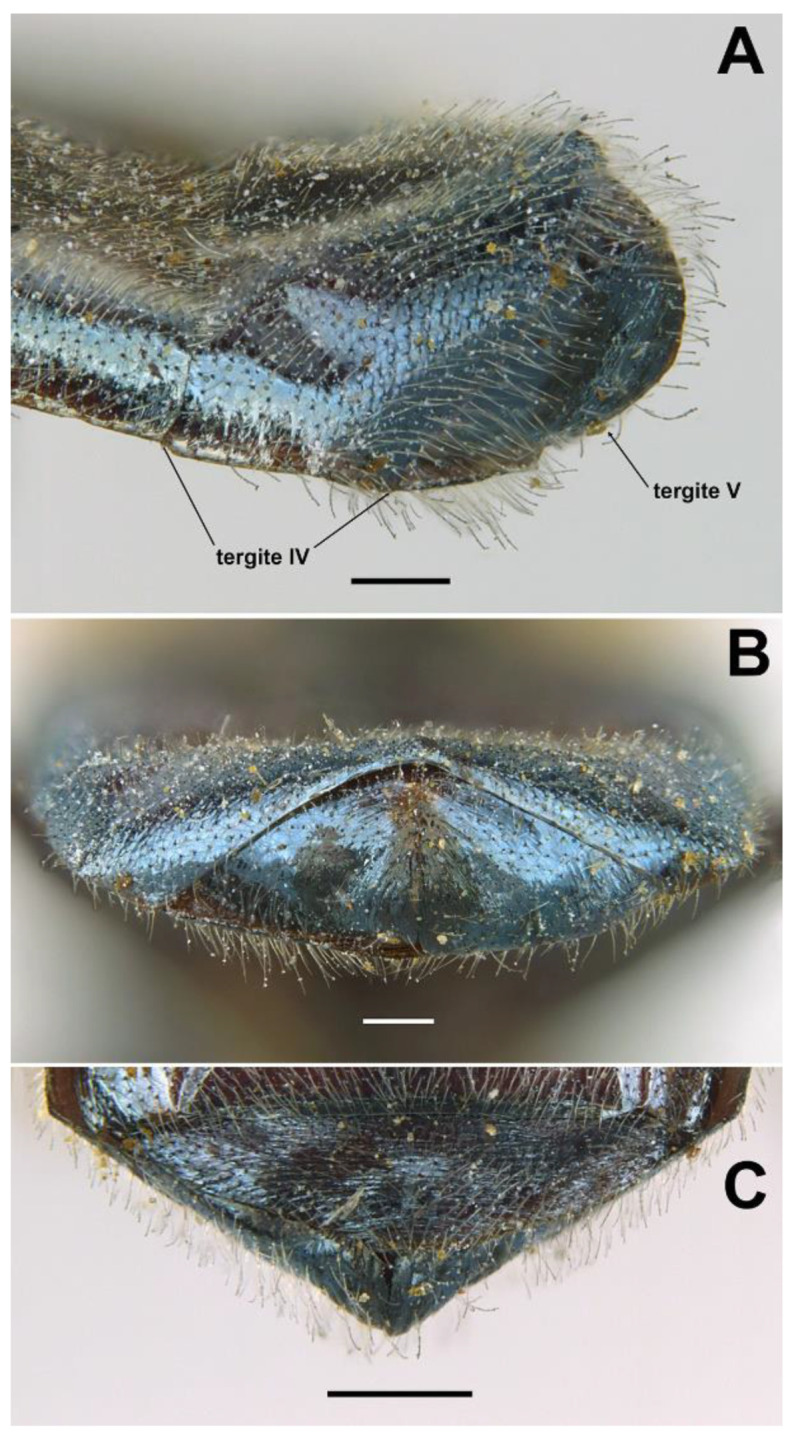
*Orthonevra plumbago*, female lectotype, abdomen apex. (**A**): Lateral view. (**B**): Dorso-posterior view. (**C**): Ventral view. Scale bars = (**A**,**B**): 0.25 mm; (**C**): 0.5 mm.

**Figure 18 insects-13-00648-f018:**
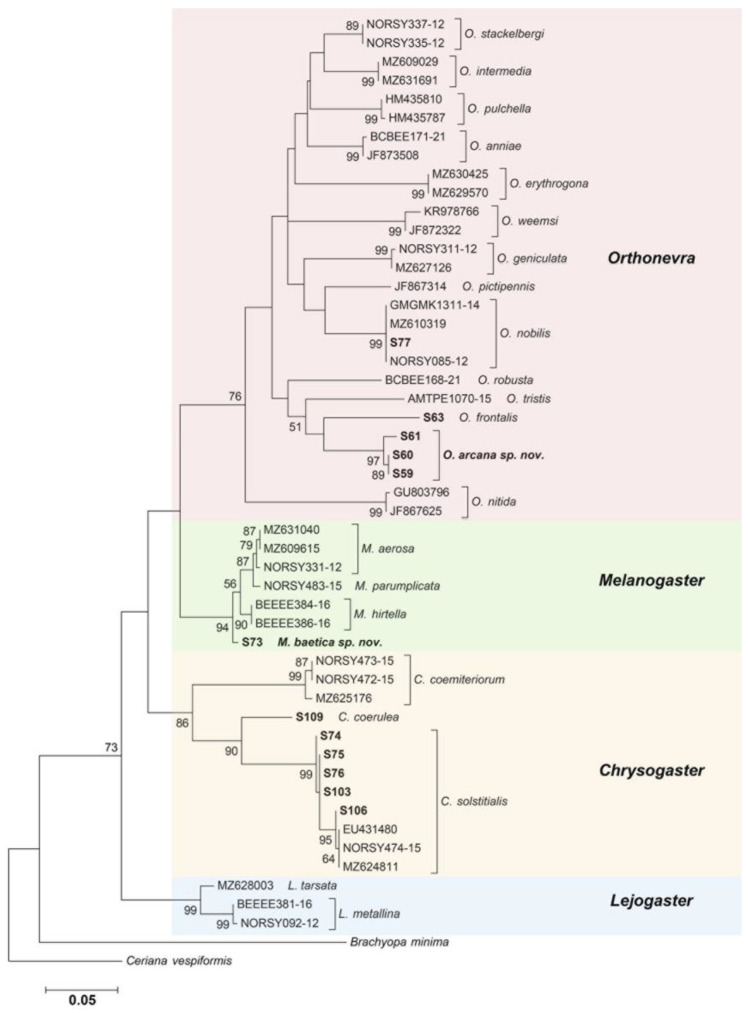
Maximum likelihood tree. DNA vouchers of specimens analysed for this work are highlighted in bold. Bootstrap values > 50 are shown near nodes. Branch lengths are measured in number of substitutions per site.

**Table 1 insects-13-00648-t001:** List of the Brachyopini species used in the molecular analyses.

Species	Life Stage	Country of Origin	DNA Voucher	Acc. Number (Genbank)	Specimen ID (BOLD)	Reference
** *Brachyopa minima* ** **Vujić and Pérez-Bañón *in* Pérez-Bañón et al. (2016)**	Larva	Greece	MZH_Y543	KU315385	GBDP31699-19	[28]
***Ceriana vespiformis* (Latreille, 1809)**	Adult	Spain	Jeff_Skevington_Specimen25788	MK972432	SYCNC169-16	[29]
***Chrysogaster coemiteriorum* (Linnaeus, 1758)**	Adult (♂)	Finland	KWi-325	MZ625176	DIFIA390-12	[30]
***C. coemiteriorum* (Linnaeus, 1758)**	Adult (♀)	Norway	NORSY472	-	NORSY472-15	[21]
***C. coemiteriorum* (Linnaeus, 1758)**	Adult (♀)	Norway	NORSY473	-	NORSY473-15	[21]
** *C. coerulea* ** **Strobl *in* Czerny and Strobl, 1909**	Adult (♀)	Spain	CEUA_S109	ON103531	-	Present work
** *C. solstitialis* ** **(Fallén, 1817)**	Adult (♀)	Norway	NORSY474	-	NORSY474-15	[21]
** *C. solstitialis* ** **(Fallén, 1817)**	-	Finland	MZH_DNA_voucher_Y583_2007	EU431480	-	[31]
** *C. solstitialis* ** **(Fallén, 1817)**	Adult (♂)	Finland	jka11-00486	MZ624811	FIDIP2612-12	[30]
** *C. solstitialis* ** **(Fallén, 1817)**	Adult (♂)	Spain	CEUA_S74	ON103525	-	Present work
** *C. solstitialis* ** **(Fallén, 1817)**	Adult (♂)	Spain	CEUA_S75	ON103526	-	Present work
** *C. solstitialis* ** **(Fallén, 1817)**	Adult (♀)	Spain	CEUA_S76	ON103527	-	Present work
** *C. solstitialis* ** **(Fallén, 1817)**	Adult (♀)	Spain	CEUA_S103	ON103529	-	Present work
** *C. solstitialis* ** **(Fallén, 1817)**	Adult (♂)	Spain	CEUA_S106	ON103530	-	Present work
** *Lejogaster metallina* ** **(Fabricius, 1777)**	Adult (♂)	Norway	NorSy92	-	NORSY092-12	[21]
** *L. metallina* ** **(Fabricius, 1777)**	Adult (♂)	United Kingdom	BMNH(E)#1731751	-	BEEEE381-16	[21]
** *L. tarsata* ** **(Meigen, 1822)**	Adult (♀)	Finland	jka07-01903	MZ628003	FIDIP2534-12	[30]
** *Melanogaster aerosa* ** **(Loew, 1843)**	Adult (♂)	Finland	MZH_HP.691	MZ631040	FIDIP4127-14	[30]
** *M. aerosa* ** **(Loew, 1843)**	Adult (♀)	Finland	MZH_HP.749	MZ609615	FIDIP955-12	[30]
** *M. aerosa* ** **(Loew, 1843)**	Adult (♂)	Norway	NorSy331	-	NORSY331-12	[21]
** *M. baetica* ** **Ricarte and Nedeljković sp. n.**	Adult (♂)	Spain	CEUA_S73	ON103524	-	Present work
** *M. hirtella* ** **(Loew, 1843)**	Adult (♀)	United Kingdom	BMNH(E)#1731754	-	BEEEE384-16	[21]
** *M. hirtella* ** **(Loew, 1843)**	Adult (♂)	United Kingdom	BMNH(E)#1731756	-	BEEEE386-16	[21]
** *M. parumplicata* ** **(Loew, 1840)**	Adult (♀)	Norway	NORSY483	-	NORSY483-15	[21]
** *Orthonevra anniae* ** **(Sedman, 1966)**	Adult (♀)	Canada	BIOUG<CAN>:10JSROW-0602	JF873508	JSDIP602-10	[32]
** *O. anniae* ** **(Sedman, 1966)**	Adult (♂)	Canada	BIOUG73902-G04	-	BCBEE171-21	[21]
** *O. arcana* ** **Ricarte and Nedeljković sp. n.**	Adult (♀)	Spain	CEUA_S59	ON103520	-	Present work
** *O. arcana* ** **Ricarte and Nedeljković sp. n.**	Adult (♂)	Spain	CEUA_S60	ON103521	-	Present work
** *O. arcana* ** **Ricarte and Nedeljković sp. n.**	Adult (♀)	Spain	CEUA_S61	ON103522	-	Present work
** *O. erythrogona* ** **(Malm, 1863)**	Adult (♂)	Finland	MZH_HP.1138	MZ629570	FIDIP3700-13	[30]
** *O. erythrogona* ** **(Malm, 1863)**	Adult (♀)	Finland	MZH_HP.1139	MZ630425	FIDIP3701-13	[30]
** *O. frontalis* ** **(Loew, 1843)**	Adult (♂)	Spain	CEUA_S63	ON103523	-	Present work
** *O. geniculata* ** **(Meigen, 1830)**	Adult (♀)	Finland	KWi-095	MZ627126	DIFIA539-12	[30]
** *O. geniculata* ** **(Meigen, 1830)**	Adult (♂)	Norway	NorSy311	-	NORSY311-12	[21]
** *O. intermedia* ** **(Lundbeck, 1916)**	Adult (♂)	Finland	MZH_HP.37	MZ631691	FIDIP037-11	[30]
** *O. intermedia* ** **(Lundbeck, 1916)**	Adult (♀)	Finland	MZH_HP.756	MZ609029	FIDIP962-12	[30]
** *O. nitida* ** **(Wiedemann, 1830)**	Adult (♀)	Canada	BIOUG<CAN>:PCPP10-0132	JF867625	BBDCN037-10	[32]
** *O. nitida* ** **(Wiedemann, 1830)**	Adult (♂)	United States	BIOUG<CAN>:09BBDIP-0534	GU803796	USDIP534-09	[21]
** *O. nobilis* ** **(Fallén, 1817)**	Adult (♀)	Finland	MZH_HP.33	MZ610319	FIDIP033-11	[30]
** *O. nobilis* ** **(Fallén, 1817)**	-	Germany	BIOUG17224-E02	-	GMGMK1311-14	[33]
** *O. nobilis* ** **(Fallén, 1817)**	Adult (♂)	Norway	NorSy85	-	NORSY085-12	[21]
** *O. nobilis* ** **(Fallén, 1817)**	Adult (♂)	Spain	CEUA_S77	ON103528	-	Present work
** *O. pictipennis* ** **(Loew, 1863)**	Adult (♀)	Canada	BIOUG<CAN>:10BBCDIP-1237	JF867314	BBDCM522-10	[32]
** *O. pulchella* ** **(Williston, 1887)**	Adult (♂)	Canada	BIOG<CAN>:09BBEDI-0456	HM435787	BBDEC456-09	[32]
** *O. pulchella* ** **(Williston, 1887)**	Adult (♀)	Canada	BIOUG<CAN>:09BBEDI-0480	HM435810	BBDEC480-09	[32]
** *O. robusta* ** **(Shannon, 1916)**	Adult (♂)	Canada	BIOUG73902-G01	-	BCBEE168-21	[21]
** *O. stackelbergi* ** **Thompson and Torp, 1982**	Adult (♀)	Norway	NorSy335	-	NORSY335-12	[21]
** *O. stackelbergi* ** **Thompson and Torp, 1982**	Adult (♂)	Norway	NorSy337	-	NORSY337-12	[21]
** *O. tristis* ** **(Loew, 1871)**	Adult (♀)	Germany	BC-ZSM-DIP-24053-C01	-	AMTPE1070-15	[21]
** *O. weemsi* ** **(Sedman, 1966)**	Adult (♂)	United States	BIOUG<CAN>:10BBDIP-0834	JF872322	DIPUS834-10	[21]
** *O. weemsi* ** **(Sedman, 1966)**	Adult (♀)	Canada	BIOUG17325-E11	KR978766	SSKJC2166-15	[32]

## Data Availability

All sequences generated in this work are available in the publicly accessible repository of GenBank (https://www.ncbi.nlm.nih.gov/genbank/).

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
