# Peer review of "Assessing the Diversity and Systematics of Brachyopini Hoverflies (Diptera: Syrphidae) in the Iberian Peninsula, Including the Descriptions of Two New Species"

_insects, 2022, doi:10.3390/insects13070648_

Round 1

Reviewer 1 Report

A very thorough work on the Brachyopini of the Iberian Peninsula. I made just a few comments on the text, hopefully to help with clarity. My only larger concern is on the new species M. baetica and its apparent very close proximity to M. hirtella and M. parumplicata, but see comment in Taxonomic notes of M. baetica in the reviewed pdf.

Reviewer 2 Report

This MS represents valuable taxonomic study on several hoverfly genera from tribe Brachyopini and enhance knowledge on Iberian fauna, as well enlarge COI data base. It is comprehensive study that includes large amount of published and new records, and even descriptions of two new species for the science. Also authors emphasized particular taxonomic issues and gave current taxonomic status of recorded species.

There are minor corrections that should be edited (all is marked in the pdf file), and after that should be considered for publishing in journal Insects.
